# Teaching with Uncertainty: Unleashing the Potential of Knowledge Distillation in Object Detection

## Abstract

Knowledge distillation (KD) has become a fundamental technique for model compression in object detection tasks. The data noise and training randomness may cause the knowledge of the teacher model to be unreliable, referred to as knowledge uncertainty. Existing methods only transfer this knowledge and could limit the student's ability to capture and understand the potential "dark knowledge". In this work, we introduce a new strategy that explicitly incorporates knowledge uncertainty, named **U**ncertainty-Driven Knowledge **E**xtraction and **T**ransfer (UET). Given that the knowledge distribution is unknown and high-dimensional in practice, we introduce a simple yet effective sampling method with Monte Carlo dropout (MC dropout) to estimate the teacher's knowledge uncertainty. Leveraging information theory, we integrate knowledge uncertainty into the conventional KD process, allowing the student model to benefit from knowledge diversity. UET is a plug-and-play method that integrates seamlessly with existing distillation techniques. We validate our approach through comprehensive experiments across various distillation strategies, detectors, and backbones. Specifically, UET achieves state-of-the-art results, with a ResNet50-based GFL detector obtaining 44.1% mAP on the COCO dataset—surpassing baseline performance by 3.9%.

## 1 Introduction

Deep learning-based object detectors have demonstrated strong performance in fundamental tasks LeCun et al. (2015); Girshick et al. (2014), but their large-scale parameters limit deployment in resource-constrained, real-world applications. Knowledge distillation (KD) Hinton et al. (2015) offers an effective model compression strategy, leveraging well-trained teacher model's dark knowledge to guide student models. In object detection, aligning the intermediate representations between teacher and student models is a common practice Cao et al. (2022); Yang et al. (2022c); Zhang & Ma (2023); Kang et al. (2021), often following the "Knowledge Extraction-Transfer (ET)" paradigm. However, a key question arises: *Is the teacher's knowledge always reliable for distillation?*

The answer might be negative. In real-world scenarios, uncertainty is intrinsic to knowledge and critically affects knowledge transfer Szulanski (2000). In object detection, uncertainty arises from noisy training data and the inherent randomness of model training. Despite careful annotation, imprecise bounding boxes introduce aleatoric uncertainty Kendall & Gal (2017), particularly in complex scenarios like occlusions. Additionally, model design and training randomness contribute to epistemic uncertainty Kendall & Gal (2017). Together, these uncertainties shape the teacher's knowledge, which is often overlooked in traditional distillation paradigms, potentially limiting the student model's ability to learn latent knowledge. To explore this further, we conducted repeated training on the COCO 2017 dataset Lin et al. (2014) using the same teacher detector Li et al. (2020), resulting in two distinct teacher models, A and B (Fig. 1 (a)). Both achieved nearly identical performance (mAP 44.9, Fig. 1 (c)). However, when distilling knowledge from these teachers to students C and D using FGD Yang et al. (2022c), although their performance was similar, their feature representations varied significantly (Fig. 1 (b)). This suggests that: (1) their feature representations differ due to aleatoric and epistemic uncertainties, which we refer to as knowledge uncertainty in this work, and (2) both teachers A and B contribute valuable knowledge. It is well known that knowledge distillation relying on multiple teachers' knowledge can offer diverse insights and has

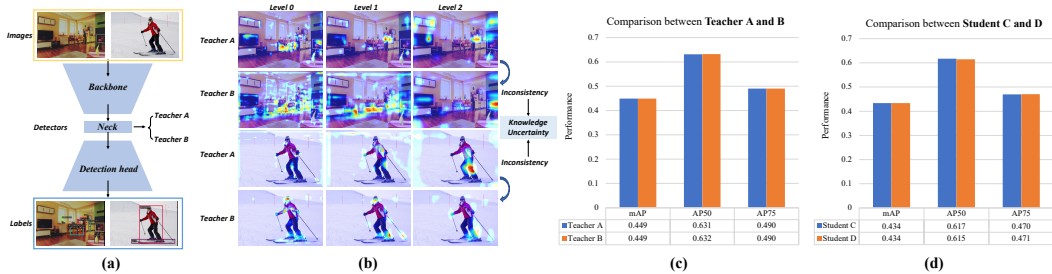

Figure 1: Why rely on a single teacher's deterministic knowledge when both teachers or even additional ones, may offer diverse insights? (a): is the representation of the training process. (b): Heatmap visualization of teacher A and B. (c): Compared with teacher A and B. (d): Compared with student C and D.

been extensively studied Son et al. (2021); Wu et al. (2021); Pham et al. (2023); Liu et al. (2020). Traditional ET paradigms align only the deterministic knowledge from one teacher, treating it as a "hard label" providing valuable knowledge, which also limits the student model's ability to learn potential "dark knowledge". While uncertainty quantification has proven beneficial in fields like segmentation Holder & Shafique (2021) and localization Yang et al. (2022b), the high-dimensional and unknown distribution of teacher knowledge makes direct quantification challenging. Ensemble models can capture uncertainty but are computationally expensive and impractical in many cases.

In this work, we propose a novel distillation paradigm, **U**ncertainty-driven Knowledge **E**xtraction and **T**ransfer (UET). Unlike the conventional ET approach, UET integrates both deterministic and uncertain knowledge based on information theory, allowing the student model to benefit from both the precision of deterministic knowledge and the diversity of uncertain knowledge. To estimate the teacher's knowledge uncertainty, we employ Monte Carlo dropout (MC dropout) Gal & Ghahramani (2016), offering a simple yet effective way to capture and integrate uncertainty into the distillation process. Importantly, UET is a plug-and-play method that integrates seamlessly with existing distillation techniques. We validate the effectiveness of UET through extensive experiments on the MS COCO dataset, applying it across various distillation strategies, detectors, and backbone architectures. When using FGD as the baseline, ResNet50-based detectors such as GFL Li et al. (2020), Faster R-CNN Ren et al. (2015), RetinaNet Lin et al. (2017b), and FCOS Tian et al. (2020) achieve mAP scores of 44.1%, 40.8%, 39.9%, and 42.9% respectively, outperforming prior state-of-the-art KD methods. In summary, the contributions of this paper can be summarized:

- We propose a new paradigm of "**U**ncertainty-Driven Knowledge **E**xtraction and **T**ransfer", for the feature-based distillation methods. This approach integrates deterministic knowledge and uncertain knowledge based on the information theory, which can effectively guide the student detector's training.

- We propose a simple yet effective method for uncertainty estimation by combining Monte Carlo dropout, enabling the capture of the teacher model's uncertainty during the knowledge distillation process. This method seamlessly integrates with existing distillation methods, delivering improvements without any additional complexity.

- We conduct extensive experiments on the COCO dataset to verify the effectiveness of the proposed paradigm across various distillation strategies, detectors, and backbone architectures, achieving SoTA performance.

## 2 RELATED WORKS

### 2.1 OBJECT DETECTION WITH KNOWLEDGE DISTILLATION

Object detection is one of the most fundamental tasks in image processing. Due to the advancements in deep learning, detectors based on convolutional neural networks (CNNs) have achieved remarkable results Li et al. (2020); Lin et al. (2017b); Tian et al. (2020); Ren et al. (2015). Additionally,

recent developments have introduced anchor-free methods (like FCOS Tian et al. (2020)) to reduce the detector's reliance on anchors. In this work, we explore the effectiveness of the proposed method across different types of detectors.

Knowledge distillation aims to transfer knowledge from a well-trained but cumbersome teacher detector to a lightweight student detector. Currently, KD in object detection can be categorized into feature-based distillation Cao et al. (2022); Yang et al. (2022c); Zhang & Ma (2023); Kang et al. (2021); Huang et al. (2023); Yang et al. (2022d); Zhu et al. (2023); De Rijk et al. (2022) and logit-based distillation Zheng et al. (2023); Wang et al. (2023); Yang et al. (2023); Zhao et al. (2022). Due to its simpler and more uniform form, feature-based knowledge distillation is popular in object detection. Presently, feature-based knowledge distillation follows the paradigm of "ET". Most works focus on the design of discriminative knowledge extraction modules (such as scale-aware knowledge Zhu et al. (2023), relationship-related knowledge Tian et al. (2020), etc.) and transfer methods (Pearson Correlation Coefficient Cao et al. (2022), SSIM De Rijk et al. (2022), etc.). In contrast to other works, we consider the uncertainty of knowledge in the teacher model and propose a novel UET paradigm for feature-based distillation. We aim to improve the existing feature distillation procedure by following UET to help the student model explore potential knowledge.

## 2.2 Uncertainty estimation

Uncertainty often arises from insufficient knowledge and data during model training, prompting the need for robust uncertainty estimation methods to quantify prediction reliability Gal & Ghahramani (2016); Hüllermeier & Waegeman (2021); Kendall & Gal (2017); Kononenko (1989); Jalonen (2012); Duncan et al. (2017); Sun et al. (2017); Milanés-Hermosilla et al. (2021); Ledda et al. (2023). Recent research has witnessed a surge in exploring uncertainty in the knowledge distillation task. For instance, the UNIX Xu et al. (2023a) proposed to reduce computation costs by combining uncertainty sampling and adaptive mixup to prioritize informative samples. Uncertainty distillation method Holder & Shafique (2021), on the other hand, aimed to quantify prediction uncertainty by training a compact model to mimic the output distribution of a large ensemble of models, enabling efficient and reliable uncertainty estimation. Similarly, the Uncertainty-aware Contrastive Distillation (UCD) Yang et al. (2022a) method attempts to alleviate catastrophic forgetting in incremental semantic segmentation by contrasting features between new and frozen models. PAD Zhang et al. (2020) proposed prime-aware adaptive distillation, which incorporates uncertainty learning to identify prime samples in distillation and adaptively emphasize their impact. Similarly, AKD Zhang et al. (2023) strived to generate inference ensemble models from a teacher model and adaptively adjust their contribution to knowledge transfer using an uncertainty-aware factor, refining the previous distillation methods for dense prediction tasks. Despite these advances, existing uncertainty estimation methods often require fine-grained modeling, and there has been limited exploration in the object detection domain. To address this limitation, we introduce knowledge uncertainty into the KD framework for object detection, guiding students to learn more potential knowledge to enhance its detection performance.

## 3 Methods

In this section, we first present the conventional ET paradigm of distillation. Next, we formulate our proposed UET paradigm, which aims to introduce knowledge uncertainty into the feature distillation process. Additionally, we provide the pseudocode of the UET paradigm in the Appendix A.4. Besides, we introduce a simple yet effective knowledge uncertainty modeling method based on the MC dropouts. Finally, we proposed a joint uncertain and deterministic knowledge distillation method in our UET paradigm. The overview of UET is represented in Figure 2.

### 3.1 KD method in ET paradigm

In object detection, we typically construct a multi-scale feature using the FPN network Lin et al. (2017a) to enhance the detector's perception of features at different scales. In feature-based knowledge distillation, knowledge transfer usually occurs on multi-scale features and follows the ET paradigm. For simplicity, we focus on single-level feature distillation in this section. Specifically, knowledge extraction methods $f_{\mathbf{E}}(\cdot)$ are first used to extract the discriminative knowledge that the

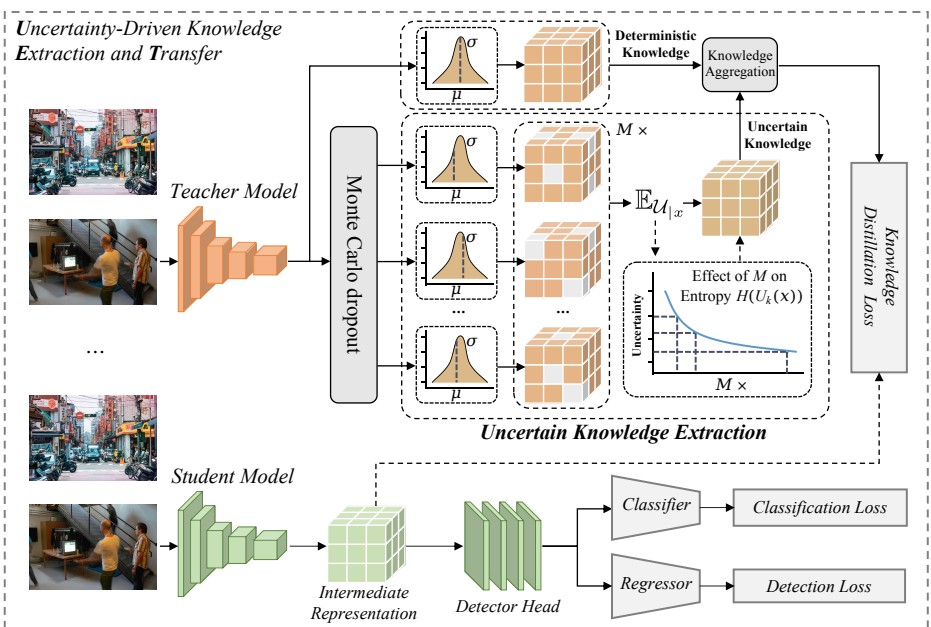

Figure 2: The overview of the proposed UET paradigm.

student model is expected to learn from the multi-scale features of the teacher model, such as the GC module Cao et al. (2023) used in the FGD method Yang et al. (2022c) for extracting pixel-wise relationships. Then, a designed constraint function $d_{\mathbf{T}}(\cdot)$ is used to minimize the difference between the discriminative knowledge of the teacher and student models, such as the SSIM De Rijk et al. (2022), to complete knowledge transfer. This process can be formalized as:

$$\arg\min_{\theta} \mathcal{L}_{\text{KD}}(\theta) = \frac{1}{N} \sum_{i} d_{\mathbf{T}}(f_{\mathbf{E}}(F^T(x_i)), f_{\mathbf{E}}(g(F^S(x_i, \theta)))), \tag{1}$$

where $x_i$ is the input image from training data, $\theta$ is the parameters of the student detector, $F^T(x)$ and $F^S(x)$ represent the FPN network's output features of the teacher and student models, respectively. Additionally, $g(x)$ denotes the adaptive function typically used to align the multi-scale features of the teacher and student models. It is worth mentioning that during the training process, only the parameters of the student model are updated through back-propagation.

## 3.2 FORMULATION OF KD WITH KNOWLEDGE UNCERTAINTY

As mentioned earlier, the inevitable uncertainty in the knowledge of the teacher model arises from data noise and the randomness of training. However, the ET paradigm overlooks the inherent uncertainty in the knowledge of the teacher model, which may limit the student's ability to learn potential "dark knowledge". To incorporate the knowledge uncertainty of the teacher model into the KD process, we assume that the teacher's knowledge, $(\mathbf{x}, \mathbf{F}^T)$, follows a distribution $\mathcal{U}$ over $\mathcal{X} \times \mathcal{Y}$, where $\mathcal{X}$ is the input instance space and $\mathcal{Y}$ represents the knowledge space. Given this assumption, the teacher's knowledge can be better represented by leveraging a sample, $\{F_j^T(\mathbf{x}) \mid j = 1, \ldots, M\}$, if available. Based on this, the KD objective can be formulated as:

$$\arg\min_{\theta} \widehat{\mathcal{L}}_{\text{KD}}(\theta) = \mathbb{E}_{\mathcal{U}_x}\{d_{\mathbf{T}}(f_{\mathbf{E}}(\frac{1}{M} \sum_{j}^{M} F_j^T(\mathbf{x})), f_{\mathbf{E}}(g(F^S(\mathbf{x}, \theta))))\}, \tag{2}$$

where $\frac{1}{M} \sum_{j}^{M} F_j^T(\mathbf{x})$ represents the empirical mean of the teacher's knowledge for input $\mathbf{x}$. By minimizing this objective, the student model incorporates the uncertainty inherent in the teacher's knowledge during optimization.

### 3.3 JOINT KNOWLEDGE DISTILLATION OF DETERMINISTIC AND UNCERTAINTY

After formulating the feature distillation with knowledge uncertainty, we can make use of the teacher's knowledge uncertainty in the training process of the student model. However, the conditional distribution of the teacher's knowledge is unknown, which results in sampling from this distribution is also challenging. Modeling the uncertainty of the deep learning model has been a topic of considerable attention Hüllermeier & Waegeman (2021); Kendall & Gal (2017). Especially, Gal & Ghahramani (2016) establishes a theoretical connection between dropout training in deep neural networks and Bayesian inference in deep Gaussian processes. More importantly, they introduced a Monte Carlo estimate method based on the dropout, referred to as MC dropout, which can realize the estimating of uncertainty. There has been a lot of work Zhang et al. (2023); Yelleni et al. (2024); Xu et al. (2023b) estimating uncertainty through MC dropout and achieving outstanding performance in specific fields. Inspired by this, we utilize MC dropout to estimate the uncertain knowledge of the teacher models. More specifically, we perform $M$-times the teacher's forward process with dropout and denote the teacher's knowledge of the $j$-th run as $Dropout_j^T(x)$. We also discuss the impact of the MC dropout on training time in Section A.5. The set $\{Dropout_j^T(x)|j=1,\ldots,M\}$ is used as a sample of the uncertain knowledge $\mathbf{F}^T$ as in Equation 2. Then, we have

$$U_K(x) \approx \frac{1}{M}\sum_{j=1}^{M} Dropout_j^T(x). \tag{3}$$

Based on Equation 3, we can incorporate the teacher model's knowledge uncertainty into the distillation process. While this uncertain knowledge introduces diversity, it focuses less on prediction performance and inevitably brings in random noise. However, the original teacher's deterministic knowledge is the result of careful training and is precision-driven. Therefore, it is crucial to guide the training of the student detector using diverse uncertain knowledge and precision-driven deterministic knowledge. Motivated by this, we combine the original teacher knowledge with the estimated uncertainty knowledge $U_K$ (similar to residual structure), enhancing the student's training process. Our proposed UET paradigm for feature distillation in object detection can be summarized as:

$$\arg\min_\theta \mathcal{L}_{\text{KD}}(\theta) = \frac{1}{N}\sum_i d_{\mathbf{T}}(f_{\mathbf{E}}(U_K(x)+F^T(x_i)), f_{\mathbf{E}}(g(F^S(x_i,\theta)))). \tag{4}$$

Although Equation 4 allows us to distill a joint of deterministic and uncertain knowledge, one question: *how to balance these two types of knowledge?* To address this, we connect this with the information entropy theory. Motivated by previous work Lin et al. (2024); Sun et al. (2022); Kendall & Gal (2017), we assume that the knowledge of the teacher model follows a Gaussian distribution: $\mathbf{F}^T \sim \mathcal{N}(\mu,\sigma^2)$. The entropy of the teacher's knowledge can be calculated as follows:

$$\begin{aligned}
H(\mathbf{F}^T) &= -\mathbb{E}[\log\mathcal{N}(\mu,\sigma^2)] \\
&= -\mathbb{E}[\log[(2\pi\sigma^2)^{-1/2}\exp(-\frac{1}{2\sigma^2}(f-\mu)^2)]] \\
&= \log(\sigma)+\frac{1}{2}\log(2\pi)+\frac{1}{2} \\
&= \frac{1}{2}\log(2\pi e\sigma^2).
\end{aligned} \tag{5}$$

We can assume that each knowledge after dropout ($Dropout_j^T(x)$) follows an independent Gaussian distribution. Therefore, the obtained uncertain knowledge $U_K(x)$ can be modeled as a Gaussian distribution:

$$U_K(x) \sim \mathcal{N}(\mu,\frac{\sigma^2}{M}). \tag{6}$$

Using Equation 5, we can calculate the entropy of the obtained uncertainty knowledge:

$$H(U_K(x)) = \frac{1}{2}\log(\frac{1}{M}2\pi e\sigma^2). \tag{7}$$

Equation 7 demonstrates that the entropy of the uncertain knowledge is inversely proportional to $M$. This implies that as $M$ increases, the entropy of the uncertain knowledge $U_K(x)$ will decrease,

Table 1: Comparison results in GFL framework on MS COCO. Black highlighted fonts represent the highest values. We set $M$ to 10, choose the **b)** as ratio strategy, and only introduce knowledge uncertainty into the teacher side in this experiment. The values highlighted in blue are those that have increased the most. * denotes our reproduced results.

| Method | Backbones | Schedule | mAP | $AP_{50}$ | $AP_{75}$ | $AP_S$ | $AP_M$ | $AP_L$ |
|---|---|---|---|---|---|---|---|---|
| GFL (T) | R101 | 24 | 44.9 | 63.1 | 49.0 | 28.0 | 49.1 | 57.2 |
| GFL (S) | R50 | 12 | 40.2 | 58.4 | 43.3 | 23.3 | 44.0 | 52.2 |
| FitNets | R50 | 12 | 40.7 (+0.5) | 58.6 (+0.2) | 44.0 (+0.7) | 23.7 (+0.4) | 44.4 (+0.4) | 53.2 (+1.0) |
| Inside GT Box | R50 | 12 | 40.7 (+0.5) | 58.6 (+0.2) | 44.2 (+0.9) | 23.1 (-0.2) | 44.5 (+0.5) | 53.5 (+1.3) |
| Defeat | R50 | 12 | 40.8 (+0.6) | 58.6 (+0.2) | 44.2 (+0.9) | 24.3 (+1.0) | 44.6 (+0.6) | 53.7 (+1.5) |
| Main Region | R50 | 12 | 41.1 (+0.9) | 58.7 (+0.3) | 44.4 (+1.1) | 24.1 (+0.8) | 44.6 (+0.6) | 53.6 (+1.4) |
| FGFI | R50 | 12 | 41.1 (+0.9) | 58.8 (+0.4) | 44.8 (+1.5) | 23.3 (+0.0) | 45.4 (+1.4) | 53.1 (+0.9) |
| GID | R50 | 12 | 41.5 (+1.3) | 59.6 (+1.2) | 45.2 (+1.9) | 24.3 (+1.0) | 45.7 (+1.7) | 53.6 (+1.4) |
| MGD* | R50 | 12 | 42.1 (+1.9) | 60.3 (+1.9) | 45.8 (+2.5) | 24.4 (+1.1) | 46.2 (+2.2) | 54.7 (+2.5) |
| SKD | R50 | 12 | 42.3 (+2.1) | 60.2 (+1.8) | 45.9 (+2.6) | 24.4 (+1.1) | 46.7 (+2.7) | 55.6 (+3.4) |
| ScaleKD | R50 | 12 | 42.5 (+2.3) | - | - | 25.9 (+2.6) | 46.2 (+2.2) | 54.6 (+2.4) |
| PKD* | R50 | 12 | 42.5 (+2.3) | 60.9 (+2.5) | 46.0 (+2.7) | 24.2 (+0.9) | 46.7 (+2.7) | 55.9 (+3.7) |
| LD | R50 | 12 | 43.0 (+2.8) | 61.6 (+3.2) | 46.6 (+3.3) | 25.5 (+2.2) | 47.0 (+3.0) | 55.8 (+3.6) |
| BCKD | R50 | 12 | 43.2 (+3.0) | 61.6 (+3.2) | 46.9 (+3.6) | 25.7 (+2.4) | 47.3 (+3.3) | 55.9 (+3.7) |
| FGD* | R50 | 12 | 43.4 (+3.2) | 61.7 (+3.3) | 47.0 (+3.7) | 26.2 (+2.9) | 47.4 (+3.4) | 56.4 (+4.2) |
| CrossKD* | R50 | 12 | 43.6 (+3.4) | 61.9 (+3.5) | 47.4 (+4.1) | 26.1 (+2.8) | 47.9 (+3.9) | 56.4 (+4.2) |
| **FGD+Ours** | R50 | 12 | **44.1** (+3.9) | **62.3** (+3.9) | **47.8** (+4.5) | **26.6** (+3.3) | **48.2** (+4.2) | **56.9** (+4.7) |

thereby reducing the knowledge diversity introduced by the uncertainty. Based on this, we only need to find the proper number of sampling times $M$ to balance the deterministic and uncertain knowledge.

Compared to the traditional ET paradigm, UET adopts the joint distillation with the uncertain and deterministic knowledge to guide the student's training process. Unlike the complex uncertainty quantification methods typically used, our approach does not require explicit quantification of knowledge uncertainty. Instead, we introduce a simple yet effective method that integrates knowledge uncertainty into the student model's training with minimal computational cost, which is almost negligible. Additionally, our method functions as a plug-and-play solution for other distillation models, enhancing the performance of the student model without additional complexity.

## 4 EXPERIMENTS

### 4.1 SETTINGS

To validate the effectiveness of our methods, we conduct extensive experiments on the MS-COCO dataset across various detectors, KDs, and different backbones. We use the 120k images of datasets to train the models and 5k val images for evaluating the experiments. We report mean Average Precision (mAP) as the evaluation metric along with AP at different scales and thresholds, including $AP_{50}$, $AP_{75}$, $AP_S$, $AP_M$, and $AP_L$. All experiments are conducted on a machine equipped with 2 NVIDIA GeForce RTX 3090 GPUs. Additionally, we select FGD Yang et al. (2022c) as our baseline, completing the "ET" process in our proposed "UET" paradigm. All experiments were implemented using the mmdetection Chen et al. (2019) and PyTorch framework. Except for setting the batch size to 8, we follow the training settings of FGD Yang et al. (2022c). In this paper, $1\times$, $2\times$, and ms respectively denote 12 epochs, 24 epochs, and multi-scale training. Unless otherwise specified, we set $M$=5 in our experiments. The first group employs a ratio strategy with an initial ratio of 0.05 and a common difference of 0.05. In these experiments, we introduce knowledge uncertainty solely on the teacher's side.

### 4.2 MAIN RESULTS

**Comparison experiments with GFL framework.** To demonstrate the superiority of our approach, we conduct comparisons with the previous SoTA KD methods on the popular GFL framework. In this experiment, we utilize a ResNet101 backbone for the GFL, trained for 24 epochs with multi-

Table 2: Results in different type framework on MS COCO.

| Method | Schedule | mAP | $AP_{50}$ | $AP_{75}$ | $AP_S$ | $AP_M$ | $AP_L$ |
|--------|----------|-----|-----------|-----------|--------|--------|--------|
| *Two-stage detectors* | | | | | | | |
| Faster R-CNN-Res101 (T) | 2× | 39.8 | 60.1 | 43.3 | 22.5 | 43.6 | 52.8 |
| Faster R-CNN-Res50 (S) | 2× | 38.4 | 59.0 | 42.0 | 21.5 | 42.1 | 50.3 |
| FitNet Romero et al. (2015) | 2× | 38.9 | 59.5 | 42.4 | 21.9 | 42.2 | 51.6 |
| FRS Zhixing et al. (2021) | 2× | 39.5 | 60.1 | 43.3 | 22.3 | 43.6 | 51.7 |
| FGD Yang et al. (2022c) | 2× | 40.5 | - | - | 22.6 | 44.7 | 53.2 |
| DiffKD Huang et al. (2023) | 2× | 40.6 | 60.9 | 43.9 | 23.0 | 44.5 | **54.0** |
| Ours+FGD | 2× | **40.8 (+2.4)** | **61.0** | **44.5** | **23.5** | **44.9** | 53.7 |
| *One-stage detectors* | | | | | | | |
| RetinaNet-Res101 (T) | 2× | 38.9 | 58.0 | 41.5 | 21.0 | 42.8 | 52.4 |
| RetinaNet-Res50 (S) | 2× | 37.4 | 56.7 | 39.6 | 20.6 | 40.7 | 49.7 |
| FitNet Romero et al. (2015) | 2× | 37.4 | 57.1 | 40.0 | 20.8 | 40.8 | 50.9 |
| FRS Zhixing et al. (2021) | 2× | 39.3 | 58.8 | 42.0 | 21.5 | 43.3 | 52.6 |
| CrossKD Wang et al. (2023) | 2× | 39.7 | 58.9 | 42.5 | **22.4** | 43.6 | 52.8 |
| FGD Yang et al. (2022c) | 2× | 39.7 | - | - | 22.0 | 43.7 | **53.6** |
| DiffKD Huang et al. (2023) | 2× | 39.7 | 58.6 | 42.1 | 21.6 | 43.8 | 53.3 |
| Ours+FGD | 2× | **39.9 (+2.5)** | **59.0** | **42.7** | 22.1 | **43.9** | 53.4 |
| *Anchor-free detectors* | | | | | | | |
| FCOS-Res101 (T) | 2×, ms | 40.8 | 60.0 | 44.0 | 24.2 | 44.3 | 52.4 |
| FCOS-Res50 (S) | 2×, ms | 38.5 | 57.7 | 41.0 | 21.9 | 42.8 | 48.6 |
| FRS Zhixing et al. (2021) | 2× | 40.9 | 60.3 | 43.6 | 25.7 | 45.2 | 51.2 |
| CrossKD Wang et al. (2023) | 2× | 41.3 | 60.6 | 44.2 | 25.1 | 45.5 | 52.4 |
| DiffKD Huang et al. (2023) | 2× | 42.4 | 61.0 | 45.8 | 26.6 | 45.9 | 54.8 |
| FGD Yang et al. (2022c) | 2× | 42.7 | - | - | **27.2** | 46.5 | **55.5** |
| Ours+FGD | 2× | **42.9 (+4.4)** | **61.6** | **46.3** | 27.1 | **46.8** | 54.7 |

scale training, as our teacher network. As for the student model, we employ a ResNet50 backbone for the GFL, trained for 12 epochs. Without resorting to any elaborate techniques, our method achieves new SoTA performance, as depicted in Table 1. Specifically, our approach achieves an mAP of 44.1%, surpassing the previous SoTA distillation methods CrossKD Wang et al. (2023) (43.6% mAP) and BCKD Yang et al. (2023) (43.2% mAP), as well as the baseline FGD Yang et al. (2022c) (43.4% mAP). Moreover, our method exhibited a notable improvement of 3.9% over the original student model (40.2% mAP).

**Comparison experiments with other detectors.** To further explore the possibility of our proposed method across different types of detectors, we conduct comparative experiments on Faster R-CNN (two-stage detector), RetinaNet (anchor-based detector), and FCOS (anchor-free detector). The experimental results, as presented in Table 2, demonstrate that our method, when incorporated on top of FGD, achieves SoTA performance across three different types of detectors. Specifically, within the Faster R-CNN framework, we attain a 40.8% mAP, marking a 2.4% improvement over the student model. Similarly, in the RetinaNet detector, we achieve a 39.9% mAP, representing a 2.5% improvement over the student model. Furthermore, within the FCOS detector, we achieve a 42.9% mAP, reflecting a 4.4% improvement over the student model.

**Comparison with other uncertainty-based methods.** To ensure a fair and comprehensive comparison, we adopt the experimental setup described in Section 4.1. Since AKD Zhang et al. (2023) is an enhanced version of PAD Zhang et al. (2020), we focus on comparing our method with AKD in this analysis. When integrated with FGD, AKD enables the student detector to achieve an mAP of 43.9. In contrast, our proposed method achieves a higher mAP of 44.1, demonstrating a 0.2 mAP improvement and a relative performance gain of 28.5% over AKD. Moreover, our method narrows the gap to the teacher model to only 0.8 mAP, which is a substantial improvement in the context of KD. This performance gain can be attributed to two key factors: a). while AKD emphasizes handling uncertain knowledge, it overlooks the deterministic knowledge from the teacher model, whereas our method explicitly integrates both deterministic and uncertain knowledge to exploit their comple-

Table 3: Performance comparison of other uncertainty-based methods.

| Method | mAP | $AP_{50}$ | $AP_{75}$ | $AP_S$ | $AP_M$ | $AP_L$ |
|---|---|---|---|---|---|---|
| FGD | 43.4 | 61.7 | 47.0 | 26.2 | 47.4 | 56.4 |
| +AKD | 43.9 | **62.3** | 47.6 | 26.5 | **48.2** | 56.8 |
| +Ours($M$=10) | **44.1** | **62.3** | **47.8** | **26.6** | **48.2** | **56.9** |

Table 4: Sensitivity analysis of the ratios strategy. The $M$ is uniformly set to 5.

| Strategy | mAP | $AP_{50}$ | $AP_{75}$ | $AP_S$ | $AP_M$ | $AP_L$ |
|---|---|---|---|---|---|---|
| *a)* | **44.0** | 62.2 | 47.8 | 27.0 | 48.3 | 56.7 |
| *b)* | **44.0** | 62.3 | 47.5 | 26.7 | 48.2 | 57.0 |
| *c)* | **44.0** | 62.4 | 47.8 | 26.9 | 48.2 | 56.9 |

Table 5: Sensitivity study of the $M$. We employ the strategy *b)* as the ratio policy and teacher's knowledge uncertainty.

| $M$ | mAP | $AP_{50}$ | $AP_{75}$ | $AP_S$ | $AP_M$ | $AP_L$ |
|---|---|---|---|---|---|---|
| 0 | 43.4 | 61.7 | 47.0 | 26.2 | 47.4 | 56.4 |
| 1 | 43.9 | 62.1 | 47.5 | 26.4 | 48.1 | 57 |
| 5 | 44.0 | 62.3 | 47.5 | 26.7 | 48.2 | 57.0 |
| 10 | **44.1** | 62.3 | 47.8 | 26.6 | 48.2 | 56.9 |
| 15 | 43.8 | 61.9 | 47.6 | 26.5 | 48.0 | 57.0 |

Table 6: Effect of knowledge sources. T: teacher's knowledge uncertainty, S: student's knowledge uncertainty, R: residual structure.

| T | S | R | mAP | $AP_{50}$ | $AP_{75}$ | $AP_S$ | $AP_M$ | $AP_L$ |
|---|---|---|---|---|---|---|---|---|
| | | | 43.4 | 61.7 | 47.0 | 26.2 | 47.4 | 56.4 |
| | ✓ | ✓ | 43.6 | 61.8 | 47.3 | 26.1 | 47.8 | 56.8 |
| ✓ | | ✓ | **44.0** | 62.2 | 47.7 | 26.8 | 48.4 | 57.0 |
| ✓ | | | 43.5 | 61.7 | 46.9 | 26.0 | 47.7 | 56.4 |
| ✓ | ✓ | ✓ | **44.0** | 62.3 | 47.5 | 26.7 | 48.2 | 57.0 |

mentary strengths. b). AKD utilizes an uncertainty-weighted loss to adaptively balance uncertain predictions during distillation, whereas our method employs a novel feature distillation mechanism that jointly captures deterministic and uncertain knowledge, enhancing the transfer of both fine-grained and global information. We also provide more comparisons in the Section A.7.

These experiments validate that following our proposed UET paradigm effectively enhances the learning potential of student detectors during the KD process. Furthermore, we demonstrate that our paradigm is adaptable to various styles of detectors.

## 4.3 ABLATION ANALYSIS

In this part, we primarily investigate the impact of dropout on student learning, such as dropout ratio and frequency. Here we adopt the GFL framework, and the experimental settings are consistent with Section 4.2. Next, we will delve into these aspects in detail.

**Sensitivity analysis of the ratios strategy.** We design three different strategies to investigate the impact of dropout ratio on the model: *a)* where $M$ groups of ratios are all fixed at 0.15; *b)* where the first group has a ratio of 0.05 with a common difference of 0.05; *c)* which builds upon *b)* by increasing with epochs, at a growth rate of 0.025 per epoch. These results, as shown in Table 4, indicate that our network is not sensitive to different ratio strategies. Particularly, when adopting strategy *c)*, after 10 epochs of training, with a dropout ratio ranging from 0.3 to 0.5, our method still maintains an mAP of 44.0%.

**Sensitivity study of the $M$.** To explore the influence of the sampling count $M$ in MC dropout on the model, we conduct comparative experiments using different $M$, as presented in Table 5. According to Section 3.3, $M$ plays a pivotal role in balancing these two types of knowledge. As $M$ increases the influence of uncertain knowledge also diminishes, reducing the diversity introduced by this uncertainty. When we perform only one dropout, uncertain knowledge, and original deterministic knowledge work together, resulting in a student model mAP of 43.9%. When increasing $M$ to 10, we observe the model's reaching 44.1% mAP, a 0.7% enhancement compared to the pure FGD method. However, as $M$ increases from 10 to 15, the influence of uncertain knowledge continues to diminish, reducing the diversity introduced by this uncertainty, which leads to the student model reaching 43.8% mAP.

**Effect of knowledge sources.** We conduct comparative experiments on introducing uncertainty from different knowledge sources. As shown in Table 6, when uncertainty is exclusively integrated into the teacher's knowledge, the mAP reaches 44.0%. Similarly, when it is introduced solely into the student knowledge, the mAP is 43.6%. In both cases, these values represent enhancements of 0.6% and 0.2% respectively compared to the pure FGD method. This suggests that considering and incorporating knowledge uncertainty into the KD process benefits the learning process of the student

Table 7: Generalization to different KDs across different detectors. * denotes our reproduced results.

| Method | Schedule | mAP | $AP_{50}$ | $AP_{75}$ | $AP_S$ | $AP_M$ | $AP_L$ |
|---|---|---|---|---|---|---|---|
| GFL-Res101 (T) | 2×, ms | 44.9 | 63.1 | 49.0 | 28.0 | 49.1 | 57.2 |
| GFL-Res50 (S) | 1× | 40.2 | 58.4 | 43.3 | 23.3 | 44.0 | 52.2 |
| LD* Zheng et al. (2023) | 1× | 41.0 (+0.8) | 58.6 | 44.2 | 23.4 | 45.0 | 53.1 |
| LD+ours | 1× | **41.2 (+1.0)** | 58.7 | 44.5 | 24.4 | 45.1 | 52.9 |
| MGD* Yang et al. (2022d) | 1× | 42.1 (+1.9) | 60.3 | 45.8 | 24.4 | 46.2 | 54.7 |
| MGD+Ours | 1× | **43.0 (+2.8)** | 61.3 | 46.4 | 26.1 | 47.1 | 55.7 |
| PKD* Cao et al. (2022) | 1× | 42.5 (+2.3) | 60.9 | 46.0 | 24.2 | 46.7 | 55.9 |
| PKD+Ours | 1× | **42.6 (+2.4)** | 60.6 | 46.1 | 23.9 | 46.7 | 55.8 |
| FGD* Yang et al. (2022c) | 1× | 43.4 (+3.2) | 61.7 | 47.0 | 26.2 | 47.4 | 56.4 |
| FGD+Ours | 1× | **44.0 (+3.8)** | 62.2 | 47.7 | 26.8 | 48.4 | 57.0 |
| RetinaNet-Res101 (T) | 2×, ms | 38.9 | 58.0 | 41.5 | 21.0 | 42.8 | 52.4 |
| RetinaNet-Res50 (S) | 1× | 37.4 | 56.7 | 39.6 | 20.6 | 40.7 | 49.7 |
| MGD* | 1× | 39.3 (+1.9) | 58.6 | 41.9 | 22.3 | 43.2 | 52.3 |
| MGD+Ours | 1× | **39.6 (+2.2)** | 58.6 | 42.5 | 22.3 | 43.8 | 52.6 |
| PKD* | 1× | 39.6 (+2.2) | 58.8 | 42.7 | 22.3 | 43.8 | 54.1 |
| PKD+Ours | 1× | **39.8 (+2.4)** | 58.8 | 42.6 | 22.2 | 43.9 | 53.9 |

model. When knowledge uncertainty is simultaneously introduced to the teacher and student, the student remains at 44.0% mAP.

**Effect of original deterministic knowledge.** We also investigate the effect of residual structures, with the results presented in Table 6. When the original knowledge isn't introduced, the student detector achieves an mAP of 43.5%, which is 0.5% lower than when it is included. This is because uncertain knowledge introduces diversity but places less emphasis on prediction accuracy. In contrast, the teacher's raw feature maps come from a carefully trained, precision-driven model, which contributes significantly to improving the student's detection performance. This finding underscores the importance of combining both deterministic and uncertain knowledge in the distillation process, revealing the synergistic effect that enhances the student model's learning.

### 4.4 GENERALIZATION TO DIFFERENT KDS

To further explore the universality of the proposed "UET" paradigm, we conduct a series of experiments by introducing the knowledge uncertainty into different KD methods across different detectors. Similarly, we employ the GFL and RetinaNet frameworks for this section, following the configurations outlined in Section 4.2. Moreover, $M$ was set to 5, with a ratio strategy of *b)* for our uncertainty introducing. Specifically, we conduct the generalization experiments on three advanced feature distillation methods at the GFL framework, (MGD Yang et al. (2022d), PKD Cao et al. (2022), and FGD Yang et al. (2022c)), as shown in Table 7. Upon introducing knowledge uncertainty, MGD, PKD, and FGD yielded improvements of 2.8%, 2.4%, and 3.8%, respectively, for the student model, surpassing their corresponding pure versions of KDs. Besides, using MGD for knowledge distillation on RetinaNet improves the student model's mAP from 37.4 to 39.3 (+1.9). Adapting MGD to the UET paradigm further enhances the performance to 39.6 (+2.2). Similarly, PKD increases the mAP to 39.6 (+2.2), which is further improved to 39.8 (+2.4) with the UET adaptation. These results confirm the applicability of the proposed paradigm across different KD situations and also suggest that incorporating knowledge uncertainty can optimize the KD process.

**Extended to logits-based distillation.** In Section 4.4, we reveal that introducing knowledge uncertainty into the feature-based distillation process would contribute to student learning. A pertinent question arises: Could the knowledge uncertainty also be applied to logits-based distillation? Compared to feature-based distillation, logit-based distillation directly transfers the output results of the teacher detector, such as classification scores and detection positions. LD Zheng et al. (2023) is one of the representative logits-based distillation methods, which designs a valuable localization region to distill the student detector in a separate distillation region manner. Following the uncertainty es-

Table 8: Classification performance with our method on ImageNet dataset. T and S mean the teacher and student, respectively.

| Method | Top-1 | Top-5 |
|---|---|---|
| ResNet-50 (T) | 76.55 | 93.06 |
| MobileNet (S) | 69.21 | 89.02 |
| MGD | 72.35 | 90.71 |
| MGD+Ours | **72.83 (+0.48)** | **91.14 (+0.43)** |

Table 9: Semantic segmentation performance with our method on Cityscapes val set. T and S mean teacher and student, respectively.

| Method | Input Size | mIoU |
|---|---|---|
| PspNet-Res101 (T) | $512 \times 1024$ | 78.34 |
| PspNet-Res18 (S) | $512 \times 512$ | 69.85 |
| MGD | $512 \times 512$ | 73.63 |
| MGD + Ours | $512 \times 512$ | **74.15 (+0.52)** |

timation manner in feature-based distillation, we introduce it in the output features of the teacher detector's FPN network and then feed them into the head to obtain predicted logits, as shown in Table 7. As expected, after introducing our method, the detector's mAP reached 41.2%, compared to the original LD's mAP of 41.0%. This result further illustrates the importance of introducing knowledge uncertainty for knowledge distillation in object detection.

### 4.5 Extended to other popular tasks

**Extended to classification tasks.** We extend the proposed UET to the classification task and validate its effectiveness on the ImageNet dataset, as shown in Table 8. Using ResNet-50 as the teacher and MobileNet as the student, our method improves both Top-1 and Top-5 accuracies based on the MGD. Specifically, the Top-1 accuracy increases from 72.35% to 72.83%, a gain of 0.48%, while the Top-5 accuracy improves from 90.71% to 91.14%, a gain of 0.43%. These results demonstrate that UET consistently enhances performance in classification, reinforcing its applicability across tasks.

**Extended to segmentation tasks.** We extend the proposed UET to the semantic segmentation task and validate its effectiveness on the Cityscapes dataset, as shown in Table 9. Specifically, following the experimental configuration of MGD, we use PspNet-Res101 as the teacher model, trained for 80k iterations with an input size of $512 \times 1024$. For the student models, we use PspNet-Res18, each trained for 40k iterations with $512 \times 512$. We set the number of samplings to 5 in this experiment. After introducing our method, the mIoU of MGD can reach 74.15 for PspNet-Res18, which is 0.52 higher than the original MGD's 73.63. These results demonstrate that our method is also applicable to semantic segmentation.

### 4.6 Extended to other backbones

**Extended to the backbone of the different types.** Inspired by these prior meaningful works Miles et al. (2024); Song et al. (2022), we further evaluate the RetinaNet detector with both heterogeneous and homogeneous backbones, including Transformer-CNN, CNN-CNN, and Transformer-Transformer configurations. Detailed results are provided in A.3.

**Extended to lightweight Backbones.** To further explore the impact of the knowledge uncertainty in lightweight Backbones, we conduct experiments using the GFL framework, and the analysis and discussion of the results are provided in A.1. We also analyze the impact of introducing knowledge uncertainty on the model's convergence in lightweight detectors in A.2.

**Visualization of detection results.** We validate the effectiveness of our proposed method through visualization evidence, and more details are listed in A.6.

### 5 Conclusions

In this paper, we investigate the significance of introducing knowledge uncertainty from teacher models in object detection distillation and its impact on the learning process of student detectors. Building upon this, we propose a novel UET general distillation paradigm, aimed at facilitating the acquisition of latent knowledge by student models, while easily being adaptable to other distillation methods. Additionally, we present a simple yet effective uncertainty estimation approach by integrating MC dropout, which seamlessly introduces uncertainty knowledge at minimal computational cost. Extensive experiments validate the effectiveness of following the UET paradigm across various types of KDs, detectors, and backbones. In summary, we demonstrate that utilizing teacher knowledge uncertainty enhances the learning capabilities of student models in the KD process.

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

# A APPENDIX

## A.1 EXTENDED TO LIGHTWEIGHT BACKBONES

Expanding to lightweight detectors presents additional challenges in KD, as the knowledge gap between teachers and students tends to be larger. Furthermore, due to the limitations in learning capacity, student models may not fully comprehend the knowledge of the teacher model. This implies that knowledge incomprehensible to these student models may affect their learning process as noise. Therefore, introducing knowledge uncertainty in the teacher model may be more crucial in the KD process for lightweight detectors. To further explore this, we conduct experiments using the GFL framework, and the results are presented in Table 10. Compared with the setting of Section 4.1, we only alternate the ResNet50 with a lightweight backbone in the student detector. As hypothesized, introducing knowledge uncertainty is crucial in lightweight detectors, leading to significant improvements. When the student model with ResNet34 as the backbone, introducing knowledge uncertainty results in a detector achieving 41.9% mAP, which is 2.2% higher than the 39.7% obtained with pure FGD.

Table 10: Quantitative results for lightweight detectors.

| Student | FGD | Ours | mAP | $AP_{50}$ | $AP_{75}$ | $AP_S$ | $AP_M$ | $AP_L$ |
|---------|-----|------|-----|-----------|-----------|--------|--------|--------|
| Res18 | | | 35.8 | 53.1 | 38.2 | 18.9 | 38.9 | 47.9 |
| | ✓ | | 33.3 (-2.5) | 49.2 | 36.0 | 20.3 | 36.1 | 42.7 |
| | ✓ | ✓ | **37.9 (+2.1)** | 54.9 | 41.0 | 21.9 | 41.5 | 49.3 |
| Res34 | | | 38.9 | 56.6 | 42.2 | 21.5 | 42.8 | 51.4 |
| | ✓ | | 39.7 (+0.8) | 57.1 | 43.0 | 23.0 | 43.7 | 51.2 |
| | ✓ | ✓ | **41.9 (+3.0)** | 59.6 | 45.3 | 24.3 | 46.0 | 54.4 |

## A.2 CONVERGENCE ANALYSIS

We also analyze the impact of introducing knowledge uncertainty on the model's convergence in lightweight detectors. As illustrated in Figure 3, when we introduce knowledge uncertainty during training, the model's convergence speed significantly improved. Particularly, with ResNet34 as the backbone, the student model following our paradigm even outperformed the baseline's performance in the third epoch. This indicates that introducing knowledge uncertainty in KD detection not only enhances student learning ability but also accelerates the convergence speed.

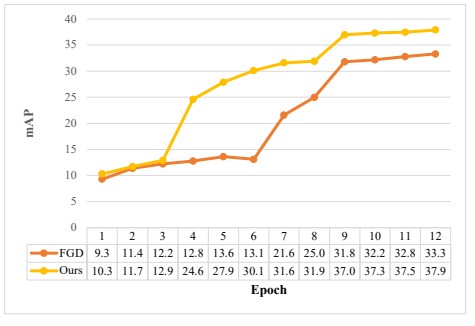

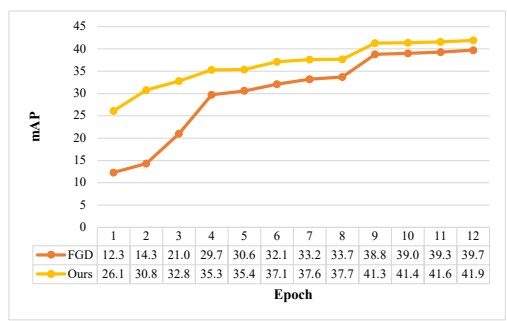

(a) The lightweight detector with ResNet18.  (b) The lightweight detector with ResNet34.

Figure 3: Convergence analysis for the lightweight detectors

## A.3 EXTENDED TO THE BACKBONE OF THE DIFFERENT TYPES

We conduct experiments on the RetinaNet detector with both heterogeneous and homogeneous backbones, and the results are presented in Table 11. Unsurprisingly, regardless of the backbone configuration being heterogeneous or homogeneous, the FGD method demonstrates stronger knowledge

transfer capabilities after following the UET paradigm. For instance, under the heterogeneous backbone setting of SwinT Liu et al. (2021), the student detector achieves 37.7% mAP after introducing knowledge uncertainty, surpassing the 37.4% obtained with pure FGD.

Considering the superior performance of Transformers in object detection, we followed prior meaningful works Miles et al. (2024); Song et al. (2022) to conduct homomorphic experiments on RetinaNet with Transformer-based architectures. Specifically, we used SwinT-tiny as the teacher backbone and SwinT-nano as the student backbone. When employing PKD for distillation, the student network achieves an mAP of 32.2. By integrating the proposed UET module, the mAP of the student model is further improved, reaching 32.5. These findings further confirm the generalization capability of UET in Transformer-based situations.

Table 11: Experiments for detectors with Heterogeneous and homogeneous backbone.

| Methods | Schedule | mAP | $AP_{50}$ | $AP_{75}$ | $AP_S$ | $AP_M$ | $AP_L$ |
|---|---|---|---|---|---|---|---|
| SwinT (T) | $1\times$ | 37.3 | 57.5 | 39.9 | 22.7 | 41.0 | 49.6 |
| Res50 (S) | $1\times$ | 36.5 | 55.4 | 39.1 | 20.4 | 40.3 | 48.1 |
| FGD | $1\times$ | 37.4 (+0.9) | 56.8 | 39.9 | 22.6 | 41.3 | 48.7 |
| FGD+Ours | $1\times$ | **37.7 (+1.2)** | 57.2 | 40.2 | 21.9 | 41.7 | 59.0 |
| SwinT-tiny (T) | $1\times$ | 37.3 | 57.5 | 39.9 | 22.7 | 41.0 | 49.6 |
| SwinT-nano (S) | $1\times$ | 31.4 | 50.5 | 32.9 | 18.0 | 34.0 | 41.5 |
| PKD | $1\times$ | 32.2 (+0.8) | 50.5 | 34.2 | 17.2 | 34.9 | 45.3 |
| PKD+Ours | $1\times$ | **32.5 (+1.1)** | 50.8 | 34.4 | 18.6 | 35.2 | 45.5 |
| Res50 (T) | $1\times$ | 36.5 | 55.4 | 39.1 | 20.4 | 40.3 | 48.1 |
| Res50 (S) | $1\times$ | 36.5 | 55.4 | 39.1 | 20.4 | 40.3 | 48.1 |
| FGD | $1\times$ | 37.4 (+0.9) | 56.7 | 39.7 | 20.6 | 40.9 | 49.0 |
| FGD+Ours | $1\times$ | **37.9 (+1.4)** | 57.3 | 40.4 | 21.2 | 41.6 | 50.1 |

### A.4 THE PSEUDO-CODE OF OUR UET PARADIGM

In comparison to the ET paradigm, we introduce the UET paradigm by incorporating knowledge uncertainty. Building upon the ET paradigm, we estimate the uncertainty in the teacher detector's knowledge using Eq. 3 and then perform knowledge transfer according to Eq. 4. Existing feature distillation methods can transform from the ET to the UET paradigm with minimal additional computational overhead (only requires a change in the **high light** parts of code).

---

**Algorithm 1** UET Paradigm

---

1: **Require:** Training data $x_{i\{i=1,...,n\}}$,
        FPN network of student detector $S_{det}$,
        FPN network of teacher detector $T_{det}$,
2: Uniformly sample a minibatch of training data $B^{(t)}$
3: **for** $x_i \in B^{(t)}$ **do**
4:     $F^T = T_{det}(x_i)$;
5:     $F^S = S_{det}(x_i)$;
6:     **procedure** UET_PARADIGM($F^T, F^S$)
7:         Estimate Uncertainty: $U_K(x) \approx \frac{1}{M} \sum_{j=1}^{M} Dropout_j(F^T(x))$
8:         KD with Knowledge uncertainty:
        $\arg\min_\theta \mathcal{L}_{KD}(\theta) = d_\mathbf{T}(f_\mathbf{E}(U_K(x) + F^T(x)), f_\mathbf{E}(g(F^S(x, \theta))))$,
9:     **end procedure**
10: **end for**

---

### A.5 IMPACT OF DROPOUT ON TRAINING TIME

Our method does not require multiple runs of the teacher network, thereby avoiding significant computational overhead. That is because multiple runs of the teacher network with dropout can

be simplified to a single run of the teacher network followed by multiple dropout applications. Moreover, we also further analyze the impact of applying dropout multiple times in our method on training time, as shown in Figure 4. Using the experimental setup described in Section 4.1 of the paper, we analyze the effect of the number of dropout applications. Without introducing any uncertainty knowledge, the training time per iteration for FGD is 0.7571 seconds. When the number of dropouts is set to 1, 5, 10, and 15, the training times for FGD are 0.7579 seconds, 0.7618 seconds, 0.7673 seconds, and 0.7720 seconds, respectively. These results indicate that our method does not significantly increase the training time.

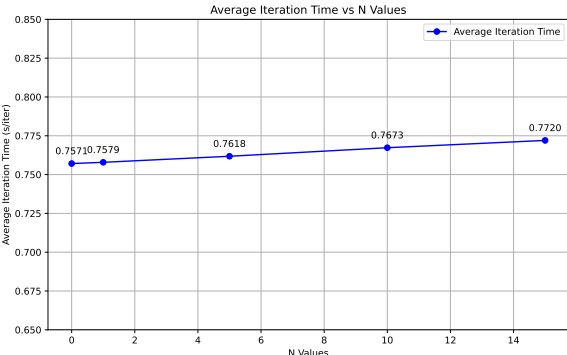

Figure 4: Impact of Dropout on Training Time.

### A.6 VISUALIZATION OF DETECTION RESULTS

We provide visual evidence to validate the effectiveness of our proposed method by presenting detection results from the val2017 set of MS COCO Lin et al. (2014). Figure 5 shows that following our UET paradigm allows FGD to outperform the original FGD in detecting more high-quality bounding boxes. This suggests that our method effectively enhances the learning potential of student detectors during the KD process.

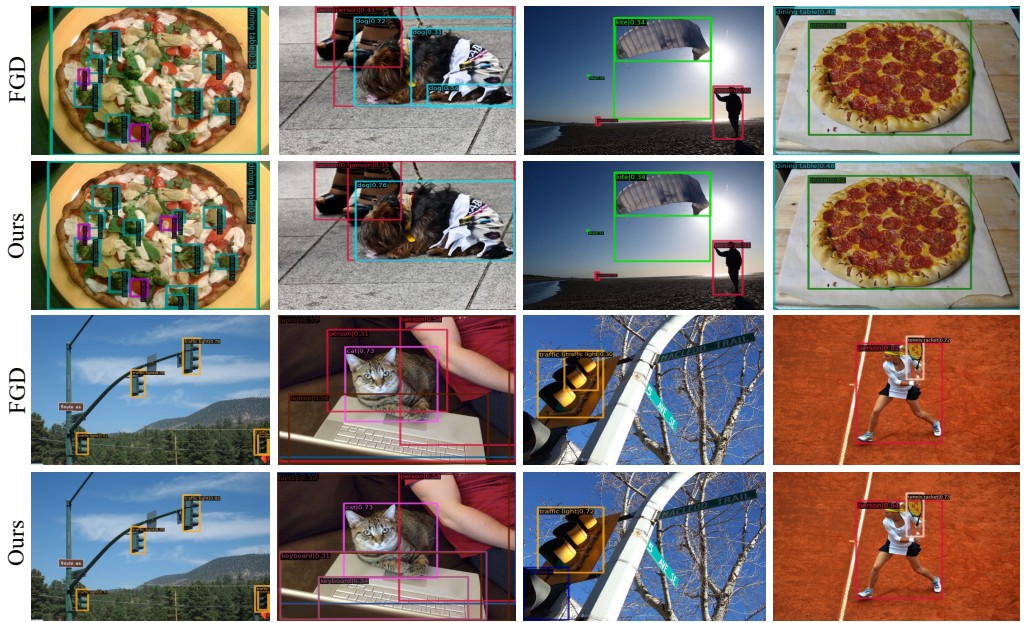

Figure 5: Visualization of detection results of Our method.

A.7 COMPARISON RESULTS WITH THE AKD IN DIFFERENT TYPE FRAMEWORK ON MS COCO.

We perform additional comparison experiments with AKD, following the setup outlined in Table 2. The results, summarized in Table 12, demonstrate the consistent superiority of our method across the Faster R-CNN and FCOS detectors. Specifically, our approach achieves an mAP of 40.8 on Faster R-CNN, surpassing AKD's 40.6, and 42.9 on FCOS, outperforming AKD's 42.7. These results highlight the effectiveness of UET in enhancing the KD process.

Table 12: Comparison Results with the AKD in different type framework on MS COCO.

| Method | Schedule | mAP | $AP_{50}$ | $AP_{75}$ | $AP_S$ | $AP_M$ | $AP_L$ |
|---|---|---|---|---|---|---|---|
| *Two-stage detectors* | | | | | | | |
| Faster R-CNN-Res101 (T) | 2× | 39.8 | 60.1 | 43.3 | 22.5 | 43.6 | 52.8 |
| Faster R-CNN-Res50 (S) | 2× | 38.4 | 59.0 | 42.0 | 21.5 | 42.1 | 50.3 |
| FGD | 2× | 40.5 | - | - | 22.6 | 44.7 | 53.2 |
| +AKD | 2× | 40.6 | 60.2 | 43.9 | 22.8 | 44.3 | 53.6 |
| +Ours | 2× | **40.8** | **61.0** | **44.5** | **23.5** | **44.9** | **53.7** |
| *One-stage detectors* | | | | | | | |
| FCOS-Res101 (T) | 2×, ms | 40.8 | 60.0 | 44.0 | 24.2 | 44.3 | 52.4 |
| FCOS-Res50 (S) | 2×, ms | 38.5 | 57.7 | 41.0 | 21.9 | 42.8 | 48.6 |
| FGD | 2× | 42.7 | - | - | **27.2** | 46.5 | **55.5** |
| +AKD | 2× | 42.7 | 61.4 | 46.1 | 26.9 | 46.5 | 54.6 |
| +Ours | 2× | **42.9** | **61.6** | **46.3** | 27.1 | **46.8** | 54.7 |

