# OpenReview forum: "Teaching with Uncertainty: Unleashing the Potential of Knowledge Distillation in Object Detection"
_ICLR.cc/2025/Conference — Submitted to ICLR 2025_

### Official Review · Reviewer_mLFV · 2024-10-30

**Soundness:** 2
**Presentation:** 2
**Contribution:** 2
**Rating:** 6
**Confidence:** 4

**Summary:**

In this work, we introduce a new strategy that explicitly incorporates knowledge uncertainty, named Uncertainty-Driven Knowledge Extraction and Transfer (UET). The author introduces a simple yet effective sampling method with Monte Carlo dropout (MC dropout) to estimate the teacher’s knowledge uncertainty, integrating knowledge uncertainty into the conventional KD process. The author estimates the proposed method on COCO dataset and achieves 44.1 mAP performance, surpassing baseline by 3.9.

**Strengths:**

- The paper introduces uncertainty estimation into object detection KD, which is kind novel in my view.
- The proposed UET achieves a SOTA performace on COCO dataset.

**Weaknesses:**

- A minor weaknesses I think is that the Monte Carlo dropout needs to sample several times, which may slow down the training process.

**Questions:**

- As shown in Tab 7, LD+UET only increase performance by 0.2 mAP from the LD only model. I am curious about why logit-based KD methods seem to less benefit from UET.
- The UET method adds teacher uncertainty knowledge into the original teacher features. Can the student also generates a uncertainty features and direct imitate the feature uncertainty, which may be more proper in my view.

---

> ### Author Response · Authors · 2024-11-21
>
> Thank you for your thorough review and constructive comments on our paper. We sincerely appreciate your recognition of our work and the valuable suggestions you provided. Based on your feedback, we have made several improvements:
>
> **R1: Effect of MC-Dropout Sampling on Training Speed**
>
> Thank you for your insightful observation. We can simulate Monte Carlo (MC) sampling by applying multiple independent dropout operations within a single forward pass. Therefore, this manner significantly avoids computational overhead, ensuring that the training time remains almost unchanged, as reflected in our experiments.
> We have analyzed the impact of MC-Dropout sampling on training speed and provided details in Appendix A.5. To summarize:
>
> Without introducing uncertainty knowledge, the training time per iteration for FGD is 0.7571 seconds.
> With MC-Dropout sampling set to 1, 5, 10, and 15, the training times are 0.7579, 0.7618, 0.7673, and 0.7720 seconds, respectively.
>
> These results demonstrate that the number of samplings has a negligible effect on training time, thanks to the efficiency of our implementation. We hope this clarifies your concern.
>
> **R2: Analyzing the Impact of UET on Logit-Based KD**
>
> Thank you for your insightful observation. The primary reason lies in the indirect influence of UET on the logit distillation process. Specifically, UET is directly applied to the multi-scale features from the FPN, which aligns well with feature-based KD methods. These methods can leverage the uncertainty knowledge directly, resulting in significant benefits. For logit-based KD, the logits are derived from the detection head, which processes features influenced by UET. This transformation may dilute the direct impact of UET.
>
> Despite this indirect effect, the combination of LD and UET still achieves a performance improvement of +0.2 mAP, indicating that UET introduces valuable guidance even in this scenario. We appreciate your observation and will clarify this distinction in the revised manuscript.
>
> **R3: Direct Imitation of Feature Uncertainty by the Student**
>
> Thank you for your insightful suggestion. Based on your valuable recommendation, we conducted additional experiments where the student directly imitates the teacher's uncertainty features under the same settings as described in Section 4.2.
>
> This manner achieved a performance of 43.5 mAP compared to the original 43.4 mAP, demonstrating that the distilled uncertainty features can benefit the student. However, this result is 0.5 mAP lower than our proposed UET (44.0 mAP).
>
> We attribute this gap to the role of the teacher's deterministic features, which are precision-driven and carefully trained. Directly imitating uncertainty alone may miss the complementary insights provided by deterministic knowledge. In contrast, UET effectively balances deterministic and uncertain knowledge, leveraging both to guide the student's learning and achieve superior performance.
>
> We appreciate your suggestion, which helped us further understand the role of uncertainty features in the distillation process.
>
> | Manner   | mAP   | $AP_{50}$ | $AP_{75}$ | $AP_S$ | $AP_M$ | $AP_L$ |
> |----------|-------|-----------|-----------|--------|--------|--------|
> | FGD      | 43.4  | 61.7      | 47.0      | 26.2   | 47.4   | 56.4   |
> | Student  | 43.5  | 61.8      | 47.2      | 26.6   | 47.5   | 56.8   |
> | **Ours** | **44.0** | **62.2** | **47.7** | **26.8** | **48.4** | **57.0** |

---

> ### Comment · Area_Chair_5mVk · 2024-11-25
>
> Dear Reviewer mLFV,
>
> Could you kindly review the rebuttal thoroughly and let us know whether the authors have adequately addressed the issues raised or if you have any further questions.
>
> Best,
>
> AC of Submission3342

---

> ### Comment · Reviewer_mLFV · 2024-11-26
>
> Thank the authors for providing new experiments and feedback. Most questions of mine have been solved after reading the rebuttal. However, I notice the reviewer pHpX mentioned that previous AKD has incorporated uncertainty learning in KD task. The newly proposed UET seems to be incremental from AKD.
>
> After briefly reading AKD, I think the differences between UET and AKD are as below:
> - UET incorporates deterministic knowledge by while AKD doesn't.
> - UET use features distillation losses while AKD uses uncertainty weighted loss.
>
> The experiments comparing AKD and UET don't show significant difference. More discussion welcomes.
>
>
>
> .

---

> > ### Author Response · Authors · 2024-11-26
> > **A different perspective on uncertainty in KD**
> >
> > We sincerely thank your comments and for recognizing the contributions of our rebuttal. We would like to address the concern regarding the comparison between UET and AKD, particularly regarding the motivation and conceptual differences in our uncertainty learning approach.
> >
> > The fundamental difference between UET and AKD lies in their conceptualization and utilization of uncertainty in KD. AKD focuses solely on addressing the diversity of uncertain knowledge from the teacher model by proposing an uncertainty-aware adaptive weighting mechanism. While this approach is valuable, it operates under the assumption that uncertain knowledge alone is sufficient for effective distillation, neglecting a critical aspect: the complementary role of deterministic knowledge.
> >
> > In contrast, UET is motivated by the insight that deterministic and uncertain knowledge are complementary in guiding the student’s learning process. Deterministic knowledge, carefully trained and precision-driven, captures the core structure and high-confidence predictions of the teacher model, which are indispensable for stabilizing and enriching the student’s training. Uncertain knowledge, on the other hand, reflects the nuanced understanding of ambiguous or diverse scenarios, providing additional flexibility. By integrating these two perspectives, UET seeks to strike a balance, leveraging both types of knowledge to achieve more robust and effective distillation.
> >
> > Key Differences Between UET and AKD:
> >
> > 1. Complementary knowledge integration vs. uncertainty weighting. AKD emphasizes handling diverse uncertain knowledge, overlooking the deterministic knowledge from the teacher model. UET explicitly integrates both deterministic and uncertain knowledge to leverage their complementary strengths, as validated by our ablation studies (Table 6 in the manuscript). For example, incorporating deterministic knowledge improves the student’s mAP from 43.5 to 44.0, demonstrating its critical importance.
> >
> > 2. Loss function design: AKD employs an uncertainty-weighted loss to adaptively weigh uncertain predictions during distillation. UET, in contrast, incorporates a novel feature distillation mechanism that jointly captures deterministic and uncertain knowledge, enhancing the transfer of fine-grained and global information.
> >
> > 3. We acknowledge the reviewer’s observation that the empirical improvements over AKD may appear modest. However, it is important to note that achieving significant performance gains in KD has become increasingly challenging due to the maturity of existing methods. If we further examine the distinctions between UET and AKD, our method achieves superior results with a more straightforward design. For instance, as shown in Table 3, UET improves the baseline mAP by 0.9, compared to AKD’s 0.7, demonstrating a relative performance gain of 28.5% over AKD. Moreover, our method narrows the gap to the teacher model to only 0.8 mAP, which is a substantial improvement in the context of KD. These results highlight the efficacy of UET and reinforce the significance of the proposed approach in advancing KD performance. We further would like to argue that UET’s strength lies in its fundamentally different perspective and motivation, which redefines how deterministic and uncertain knowledge can work together in KD. The introduction of complementary knowledge integration is not merely incremental; it represents a new way of approaching uncertainty in KD, which has been empirically validated through our ablation studies and performance comparisons.
> >
> > We hope this clarifies the key distinctions between UET and AKD and underscores the conceptual novelty and effectiveness of our approach. We are looking forward to further discussion and appreciate your constructive feedback.

---

> ### Comment · Reviewer_mLFV · 2024-11-26
>
> I sincerely appreciate the quick response from the authors. After carefully reading the response, my concern about the difference between UET and AKD has been partly solved. I will keep the original score and recommend the authors add the discussion in the main paper.

---

> > ### Author Response · Authors · 2024-11-26
> > **Authors response**
> >
> > We sincerely appreciate your recognition and valuable feedback. We will incorporate the discussion from our exchange into Section 4.2 of the main paper, as suggested. Thank you for helping us refine our motivation and discussion, making our work more comprehensive and robust. Your thoughtful input has been instrumental in improving our paper, and we are deeply grateful for your support.

---

### Official Review · Reviewer_kB7D · 2024-10-31

**Soundness:** 3
**Presentation:** 3
**Contribution:** 2
**Rating:** 6
**Confidence:** 4

**Summary:**

On this paper, the authors develop a new knowledge-distillation method targeted at the task of object detection. The method works by measuring the uncertainty of the teacher (using MC-dropout) before passing the knowledge to the student, with the idea that this should reduce the amount of noise passed to the student. The method is quite general (it works in both feature and logit distillation, and is trivially extended to classification and segmentation tasks) and is measured in MS-COCO using several convolutional-based backbones, showing that it outperforms other baselines.

**Strengths:**

I think the paper has the following strengths:

1) The idea is extremely simple, and seems to work okay in practice. In general, I tend to like papers that find gaps in literature, and are obvious in hindsight. So, while it can look A (knowledge distillation) + B (MC-dropout) = C (better results), I actually think that this is a strength rather than weakness of the paper, and are quite happy to see that at times simple ideas can outperform more complex ones.

2) The paper is relatively well written. There are some concerns about the writing (see the weaknesses section), but in general after reading the paper, I think that I understood it quite well, and would be able to reproduce it from scratch without many issues.

3) The paper for most part outperforms the other methods in several detectors, using generalized focal loss, Faster-RCNN, Retina-Net (focal loss) and FCOS, with a ResNet101 teacher and ResNet-50 student, showing a relative completeness of results in both one-stage and two-stage detectors. Furthermore, the authors show results with smaller backbones (Res18/Res-34) in Appendix, and results in classification and segmentation at the end of the paper.

**Weaknesses:**

I think the paper has the following weaknesses:

1) While the paper is relatively well-written, it still has some writing issues:

a) I feel like the writing in the Method part is slightly obfuscated and gives the feeling that the method is more complex than it is. For example, I find the equations 2 and 3 quite superflous and the writer could have directly gone to equation 4 with minimal changes. I think that the authors should be proud of their simple idea that works well, instead of making it deliberately look more complex.

b) The big claim in the first page 'Why rely on a single teacher's deterministic knowledge when both teachers or even additional ones, may offer diverse insights?' while completely true, is something that is well known in the distillation literature and has been extensively studied in the past. The authors should cite at this sentence previous works:

[A] Son et al., Densely Guided Knowledge Distillation using Multiple Teacher Assistants, ICCV 2021.

[B] Wu et al., One Teacher is Enough?Pre-trained Language Model Distillation from Multiple Teachers, ACL 2021

[C] Pham et al., Collaborative Multi-Teacher Knowledge Distillation for Learning Low Bit-width Deep Neural Networks, WACV 2023

[D] Liu et al., Adaptive Multi-Teacher Multi-level Knowledge Distillation, Neurocomputing 2020

The papers are sufficiently different to this one (ensembles instead of MC-dropout) so putting them in context of this work would strengthen, instead of weakening the paper.

2) The results while at the first look might look strong, there are a few inconsistencies there.

a) In Table 1, the results of CrossKD are not those from the paper, but results reproduced by the authors. It is unclear why the authors have done so, and when you look in the paper, the results are higher than those provided by the authors.

Method mAP AP50 AP75 APS APM APL

CrossKD_authors 43.6 61.9 47.4 26.1 47.9 56.4

CrossKD_paper 43.9 62.0 47.7 26.4 48.5 56.9

UET 44.1 62.3 47.8 26.6 48.2 56.9

As can be seen, the performance improvement from CrossKD (results reported in the paper) is marginal, only 0.2 percentage points in mAP score. Which makes the results of this paper look less impressive than what is reported.

b) Similarly, the results of FGD method reported in Table 2 under FCOS setting are not consistent with those in the paper.

Method mAP APS APM APL

FGD_authors 42.1 27.0 46.0 54.6

FGD_paper 42.7 27.2 46.5 55.5

UET 42.9 27.1 46.8 54.7

In fact, as can be seen, there is next to no improvement in this method when compared to official results of the FGD.

3) Lack of comparisons with Transformer models. While I appreciate the comprehensive evaluation done by the authors, I must say that I was disappointed to not see any Transformer-based method in this paper. Considering that Transformers are the de-facto state-of-the-art methods in virtually any machine learning task, it would have been interesting to see how this method works in Transformer setting. Some recent papers that do KD in Transformer settings:

[E] Miles et al., VkD : Improving Knowledge Distillation using Orthogonal Projections, CVPR 2024

[F] Song et al., Vidt: An efficient and effective fully transformer-based object detector (under Token matching setting), ICLR 2022

4) Lack of reporting in calibration. The entire idea of doing distillation adjusted for noise is to also ensure that the results of the detector are more reliable, not only better. To do so, the authors should have provided calibration results, where they show that their method is more precise than the others. Such an analysis would strengthen the paper and potentially offer new insights.

5) Timing performance. While discussed in Appendix A.5, it is completely unclear to me. If the authors are doing 5, 10 and 15 forward passes, how is it possible that the training time almost does not change at all. I know that backprop of the student takes more time than the forward pass of the teacher, but if you are doing k forward steps that should accumulate and increase the timing (the student will be waiting for the forward passes before the 'teacher knowledge' is passed to it).

Furthermore, it was unclear to me if the authors are doing MC-dropout in the student during inference, and if so, how much does that effect the timing.

**Questions:**

I have the following questions which the authors should ideally give a response on:

1) Please justify why the results provided in this paper are different to those reported in the original paper (weakness 2)

2) Would it be possible to have some comparison in Transformer models? (weakness 3)

3) Please provide calibration plots/results (weakness 4) and timing performance (weakness 5).

4) Please adjust the citation discussion with regards to weakness 1.

I like the idea of the paper, but I feel that the paper needs to address these issues before it is ready for acceptance. I am initially scoring the paper as (5) but will improve the score providing that the authors give satisfactory responses in my points.

---

> ### Author Response · Authors · 2024-11-21
>
> Thank you for your detailed review and valuable suggestions regarding our work. Below, we address each of your concerns and suggestions in detail.
>
> **R1 Addressing Writing Issues and Clarifications**
>
> a) Simplifying the Method Section
>
> We appreciate the suggestion to streamline Equations 2 and 3. In the revised manuscript, we have removed these equations and directly derived Equation 4. This adjustment improves clarity and ensures the method’s simplicity is effectively conveyed.
>
> b) Refining the Introduction and Adding Citations
>
> We agree that the claim on leveraging diverse teacher knowledge should be contextualized within existing literature [1-4]. We have revised the sentence in the introduction and added citations to the suggested works [1-4].
>
> Thank you again for your valuable feedback, which has significantly improved the clarity and positioning of our manuscript.
>
> [1] Son et al., Densely Guided Knowledge Distillation using Multiple Teacher Assistants, ICCV 2021.
>
> [2] Wu et al., One Teacher is Enough? Pre-trained Language Model Distillation from Multiple Teachers, ACL 2021.
>
>
> [3] Pham et al., Collaborative Multi-Teacher Knowledge Distillation for Learning Low Bit-width Deep Neural Networks, WACV 2023.
>
> [4] Liu et al., Adaptive Multi-Teacher Multi-level Knowledge Distillation, Neurocomputing 2020.
>
>
> **R2: Addressing Result Inconsistencies**
>
> We sincerely thank the reviewer for pointing out the inconsistencies in the reported results. Below, we clarify these issues:
>
> a) CrossKD Results.
>
> The reported 43.9 mAP in the CrossKD paper is achieved by combining CrossKD with PKD. Specifically, in the original CrossKD paper (Table 7), they report 43.7 mAP for pure CrossKD and 43.9 mAP for CrossKD + PKD. Additionally, we observed that in other tables, such as Table 8 and Table 9, the authors report results only for pure CrossKD.
>
> To ensure a consistent comparison and isolate the specific contribution of CrossKD itself, we reproduced its results without PKD. In our experiments, we report a reproduced mAP of 43.6, which closely matches the original CrossKD's reported 43.7 mAP. This consistency demonstrates the validity of our reproduction and provides a fair basis for comparison.
>
> b) FGD Results.
>
> We sincerely apologize for this oversight. We have corrected the reported performance of FGD under the FCOS setting.
>
> As you correctly pointed out, UET does not show a particularly significant improvement in the FCOS setting. This can be attributed to the following reasons: 1) Diminishing Returns Effect: Similar phenomena have been observed in DiffKD [5]. For instance, in Table 4 of the DiffKD paper, adopting an FGD-inspired attention-based MSE loss resulted in only a 0.1-point improvement (42.4 vs. 42.5 mAP). 2) Strong Baseline Performance of FGD: FGD already achieves high performance on FCOS, with a mAP of 42.7, surpassing DiffKD's 42.4 mAP. Even under such a strong baseline, our UET still brings an additional 0.2-point improvement (42.7 to 42.9 mAP), which is notable.
> In contrast, in the GFL setting, FGD achieves a mAP of 43.4, which is 0.2 points lower than the state-of-the-art CrossKD (43.6 mAP) and 0.5 points lower than CrossKD + PKD (43.9 mAP). By incorporating UET, FGD reaches a state-of-the-art performance of 44.1 mAP. These results further demonstrate the effectiveness of UET in knowledge distillation.
>
> Thank you for pointing this out, and we hope this clarification addresses your concern.
>
> [5] Huang T, Zhang Y, Zheng M, et al. Knowledge diffusion for distillation[J]. Advances in Neural Information Processing Systems, 2024, 36.

---

> > ### Author Response · Authors · 2024-11-21
> >
> > **R3: On Comparisons with Transformer-Based Models**
> >
> > Thank you for your thoughtful feedback and for highlighting the importance of evaluating our method in the context of Transformer-based models, given their prominence in state-of-the-art machine learning tasks.
> >
> > a) Heterogeneous Backbone Experiments
> >
> > To clarify, we have already conducted distillation experiments using Swin Transformer-Tiny as the teacher backbone and ResNet-50 as the student backbone, as shown in Table 11 of our paper. The results demonstrate that our proposed UET performs effectively even in Transformer-to-CNN heterogeneous configurations (FGD: 37.4 vs.
> > FGD+UET: 37.7).
> >
> > b) Additional Experiments on Transformer-to-Transformer Distillation
> >
> > We appreciate the insightful references [6-7] you provided and have extended our experiments to include Transformer-to-Transformer configurations. Specifically, we used Swin Transformer-tiny as the teacher backbone and Swin Transformer-nano as the student backbone. The results are shown in follows table. When employing PKD for distillation, the student network achieves an mAP of 32.2. By integrating the proposed UET module, the mAP of the student model is further improved, reaching 32.5. These findings further confirm the generalization capability of UET in Transformer-based settings.
> >
> > **Table**: Experiments for detectors with transformer backbone on the RetinaNet detector.
> >
> > | Methods                       | Schedule | mAP            | $AP_{50}$ | $AP_{75}$ | $AP_S$ | $AP_M$ | $AP_L$ |
> > |-------------------------------|----------|----------------|-----------|-----------|--------|--------|--------|
> > | SwinT-tiny (T)               | 1×       | 37.3           | 57.5      | 39.9      | 22.7   | 41.0   | 49.6   |
> > | SwinT-nano (S)               | 1×       | 31.4           | 50.5      | 32.9      | 18.0   | 34.0   | 41.5   |
> > | PKD                          | 1×       | 32.2 (+0.8)    | 50.5      | 34.2      | 17.2   | 34.9   | 45.3   |
> > | **PKD+Ours**                 | 1×       | **32.5 (+1.1)** | 50.8      | 34.4      | 18.6   | 35.2   | 45.5   |
> >
> > [6] Miles et al., VkD : Improving Knowledge Distillation using Orthogonal Projections, CVPR 2024
> >
> > [7] Song et al., Vidt: An efficient and effective fully transformer-based object detector (under Token matching setting), ICLR 2022
> >
> > **R4: Calibration Analysis**
> >
> > Thank you for highlighting the importance of calibration in evaluating the reliability of the detector. In response to your suggestion, we performed additional calibration analysis using Expected Calibration Error (ECE) and Brier Score, alongside mAP, to comprehensively assess both prediction reliability and detection performance.
> >
> > The results, summarized in the table below, show that our method achieves improved calibration (lower ECE and Brier Score) while significantly enhancing detection performance (mAP: 44.0 vs. 43.4). These findings demonstrate that our approach not only improves accuracy but also provides more reliable predictions, achieving a better balance between performance and calibration compared to FGD.
> >
> > We believe these results provide valuable insights into how UET improves both prediction reliability and accuracy, further emphasizing the robustness and practicality of our method. Thank you again for your suggestion, which helped us strengthen our evaluation.
> >
> > **Table**: Calibration Analysis Results
> >
> > | Method | ECE (%) | Brier Score (%) | mAP (%) |
> > |--------|---------|-----------------|---------|
> > | FGD    | 8.92    | 17.73           | 43.4    |
> > | Ours   | 8.91    | 17.66           | 44.0    |
> >
> > **R5: Clarifications on Timing Performance and MC-Dropout**
> >
> > Thank you for your insightful comments. We address your concerns as follows:
> >
> > a) On Training Time and Forward Passes.
> >
> > To clarify, our implementation does not perform multiple forward passes during training. Instead, we simulate Monte Carlo (MC) sampling by applying multiple independent dropout operations within a single forward pass. This significantly reduces computational overhead, ensuring that the training time remains almost unchanged, as reflected in our experiments (Appendix A.5). Specifically, without introducing uncertainty knowledge, the training time per iteration for FGD is 0.7571 seconds.
> > With MC-Dropout sampling set to 1, 5, 10, and 15, the training times are 0.7579, 0.7618, 0.7673, and 0.7720 seconds, respectively. We also revised the manuscript to make this implementation detail more explicit.
> >
> > b) On MC-Dropout During Inference.
> >
> > MC-dropout is not applied to the student model during inference. We have explicitly clarified this in the Methods section to avoid any potential misunderstandings.

---

> > > ### Comment · Reviewer_kB7D · 2024-11-25
> > > **Thanks for the rebuttal**
> > >
> > > I thank the authors for their rebuttal, clarification and the experiments. Most of my concerns have been addresed.
> > >
> > > I also appreciate the authors updating their manuscript.
> > >
> > > Thus, I increase my score to 6.

---

> > > > ### Author Response · Authors · 2024-11-26
> > > >
> > > > Thank you for your thoughtful response and for recognizing our efforts to improve the paper.
> > > >
> > > > Your invaluable feedback has been instrumental in refining our work, especially in areas such as writing, comparisons with prior works, result consistency, and additional analyses. These insights have significantly enhanced the overall quality of our paper.
> > > >
> > > > We sincerely appreciate your time, effort, and constructive comments throughout the review process. Your guidance has been crucial in strengthening our work.

---

> ### Comment · Area_Chair_5mVk · 2024-11-25
>
> Dear Reviewer kB7D,
>
> Could you kindly review the rebuttal thoroughly and let us know whether the authors have adequately addressed the issues raised or if you have any further questions.
>
> Best,
>
> AC of Submission3342

---

### Official Review · Reviewer_pHpX · 2024-11-02

**Soundness:** 2
**Presentation:** 2
**Contribution:** 2
**Rating:** 5
**Confidence:** 4

**Summary:**

This paper presents a method (UET) for improving knowledge distillation on object detection. UET leverages the MC dropout to estimate the teacher’s knowledge uncertainty, which allows the student to distill diverse knowledge from the teacher. Comprehensive experiments demonstrate the effectiveness of UET.

**Strengths:**

1.The motivation is convincing, i.e., multiple teachers can provide more diverse and informative supervision to the student.
2.UET can be seamlessly integrated with the existing KD methods and achieve a new state-of-the-art result with FGD [1] on COCO.
[1]. Focal and Global Knowledge Distillation for Detectors, 2022, CVPR.
3.	UET is successfully extended into classification and semantic segmentation tasks.

**Weaknesses:**

1.The idea is similar to [1], weakening the overall novelty of this paper, and the major difference with [1] is that the UET also included the original teacher in the distillation process. However, an ablative study on w/ and w/o the original teacher is not given, and the direct comparison with [1] is limited (only Table 3).
[1] Avatar Knowledge Distillation: Self-ensemble Teacher Paradigm with Uncertainty, 2023, MM.
2.	The superior results in Table 1 and Table 2 are obtained with FGD, which is a strong approach in distilling detectors. Therefore, the comparison with the SOTA KD methods seems to be unfair.
3.	The generalization of the proposed method is not fully evaluated. It is expected to embed UET to multiple detectors across different KD methods.
4.	 In Table 6, what is the student’s knowledge uncertainty (denoted by S). I cannot find relevant statements in the Methods section. Would the authors clarify this.

**Questions:**

See weakness.

---

> ### Author Response · Authors · 2024-11-21
>
> Thank you for your detailed review and valuable suggestions regarding our work. Below, we address each of your concerns and suggestions in detail.
>
> **R1: Compared with the AKD**
>
> 1). Formally, AKD [1] may appear similar to our approach. They propose an uncertainty-aware adaptive weighting mechanism to mitigate the interference caused by diverse knowledge effectively. However, their method focuses solely on diverse knowledge while completely neglecting the original deterministic knowledge. Deterministic knowledge from the teacher, being carefully trained and precision-driven, plays a critical role in the knowledge distillation process. As demonstrated in Table 2 of the paper, the student network achieves 43.5 mAP without the inclusion of the teacher's original deterministic knowledge. By incorporating it, the performance significantly improves to 44.0 mAP. In contrast, our method highlights the complementary roles of deterministic and uncertain knowledge, demonstrating how their joint distillation provides more effective guidance for the student’s training process.
>
> 2). We would like to clarify that Table 6 in our manuscript already includes an ablation study analyzing the impact of incorporating the original teacher's deterministic knowledge. Specifically, the column labeled "R" illustrates that excluding the original teacher results in a student mAP of 43.5, whereas including it improves performance to 44.0. These results underscore the importance of integrating deterministic knowledge alongside uncertain knowledge, validating our approach to joint distillation.
>
> 3). We have conducted further comparison experiments with AKD by following the setup in Table II. The results show that our method achieves superior performance across two detector types:
> Faster R-CNN (Ours: 40.8 vs. AKD: 40.6) and
> FCOS (Ours: 42.9 vs. AKD: 42.7).
> These findings further demonstrate the effectiveness of the UET in the KD process. The carefully trained, precision-driven deterministic knowledge, remains indispensable for knowledge distillation.
>
> | Method                          | Schedule      | mAP   | $AP_{50}$ | $AP_{75}$ | $AP_S$ | $AP_M$ | $AP_L$ |
> |---------------------------------|---------------|-------|-----------|-----------|--------|--------|--------|
> | **Two-stage detectors**         |               |       |           |           |        |        |        |
> | Faster R-CNN-Res101 (T)         | 2$\times$     | 39.8  | 60.1      | 43.3      | 22.5   | 43.6   | 52.8   |
> | Faster R-CNN-Res50 (S)          | 2$\times$     | 38.4  | 59.0      | 42.0      | 21.5   | 42.1   | 50.3   |
> | FGD                             | 2$\times$     | 40.5  | -         | -         | 22.6   | 44.7   | 53.2   |
> | +AKD                            | 2$\times$     | 40.6  | 60.2      | 43.9      | 22.8   | 44.3   | 53.6   |
> | **+Ours**                       | 2$\times$     | **40.8** | **61.0** | **44.5** | **23.5** | **44.9** | **53.7** |
> | **One-stage detectors**         |               |       |           |           |        |        |        |
> | FCOS-Res101 (T)                 | 2$\times$, ms | 40.8  | 60.0      | 44.0      | 24.2   | 44.3   | 52.4   |
> | FCOS-Res50 (S)                  | 2$\times$, ms | 38.5  | 57.7      | 41.0      | 21.9   | 42.8   | 48.6   |
> | FGD                             | 2$\times$     | 42.7  | -         | -         | **27.2** | 46.5   | **55.5** |
> | +AKD                            | 2$\times$     | 42.7  | 61.4      | 46.1      | 26.9   | 46.5   | 54.6   |
> | **+Ours**                       | 2$\times$     | **42.9** | **61.6** | **46.3** | 27.1   | **46.8** | 54.7   |
>
> [1] Zhang Y, Chen W, Lu Y, et al. Avatar knowledge distillation: self-ensemble teacher paradigm with uncertainty[C]//Proceedings of the 31st ACM International Conference on Multimedia. 2023: 5272-5280.

---

> > ### Author Response · Authors · 2024-11-21
> >
> > **R2: Clarifying the Role of UET**
> >
> > Thank you for your valuable feedback. We would like to address your concerns as follows:
> >
> > 1). Generalization Across KD Methods and Detectors
> >
> > While FGD is indeed a strong baseline for distillation, the consistent improvements achieved with UET across various KD methods demonstrate that our approach is not limited to FGD but provides complementary benefits to a wide range of frameworks. To showcase the generalizability of UET, we conducted experiments using the GFL detector with four distinct KD methods (e.g., FGD, PKD, MGD, and LD), as presented in Table 7. Additionally, based on your suggestion, we extended our analysis to include experiments on the RetinaNet detector, as discussed in R1.3. These results consistently show that UET enhances the performance of various KD methods. These findings further reinforce that UET is not tied to any specific KD framework or detector, highlighting its broad applicability and complementary nature.
> >
> > 2). Complementary Role of UET
> >
> > The primary objective of UET is not to compete directly with existing KD methods but to act as a complementary enhancement. While methods like FGD focus on deterministic knowledge distillation, they often overlook the inherent uncertainty in teacher knowledge, which serves as a valuable additional source of guidance for the student model. UET bridges this gap by introducing uncertainty as a novel dimension of knowledge in the distillation process. By integrating deterministic and uncertain knowledge, UET enables existing KD methods to unlock their full potential.
> >
> > We greatly appreciate your feedback, as it allowed us to clarify the positioning of UET and its role in complementing existing KD methods. These insights have strengthened the focus and presentation of our work.
> >
> > **R3: Generalization of UET Across Detectors and KD Methods**
> >
> > To address your concern, we provide additional experimental evidence demonstrating UET's effectiveness across different detectors and KD methods.
> >
> > Specifically, we added experiments embedding UET into multiple KD methods (MGD and PKD) on the RetinaNet detector. The results are presented in the following Table. When using MGD for deterministic knowledge distillation, the student model achieves a performance of 39.3 mAP, representing a 1.9 improvement over the original student model. By adapting MGD to follow the UET paradigm, the student model's performance further increases to 39.6 mAP, achieving a total improvement of 2.2 points over the original student.
> > Similarly, with PKD, the student model achieves 39.6 mAP, which is a 2.2 improvement over the original student. When PKD is adapted to the UET paradigm, the student model's performance further improves to 39.8 mAP, resulting in a total improvement of 2.4 points over the original student.
> >
> > These extensive experiments demonstrate that UET is a versatile and robust framework, capable of improving various KD methods across multiple detectors and configurations.
> >
> >
> > | Method                 | Schedule      | mAP           | $AP_{50}$ | $AP_{75}$ | $AP_S$ | $AP_M$ | $AP_L$ |
> > |------------------------|---------------|---------------|-----------|-----------|--------|--------|--------|
> > | RetinaNet-Res101 (T)   | 2$\times$, ms | 38.9          | 58.0      | 41.5      | 21.0   | 42.8   | 52.4   |
> > | RetinaNet-Res50 (S)    | 1$\times$     | 37.4          | 56.7      | 39.6      | 20.6   | 40.7   | 49.7   |
> > | MGD*                   | 1$\times$     | 39.3 (+1.9)   | 58.6      | 41.9      | 22.3   | 43.2   | 52.3   |
> > | **MGD+Ours**           | 1$\times$     | **39.6 (+2.2)** | 58.6      | 42.5      | 22.3   | 43.8   | 52.6   |
> > | PKD*                   | 1$\times$     | 39.6 (+2.2)   | 58.8      | 42.7      | 22.3   | 43.8   | 54.1   |
> > | **PKD+Ours**           | 1$\times$     | **39.8 (+2.4)** | 58.8      | 42.6      | 22.2   | 43.9   | 53.9   |
> >
> > **R4: Clarification of the Student's Knowledge Uncertainty (S)**
> >
> > We sincerely thank the reviewer for pointing out the need for clarification regarding the student's knowledge uncertainty (S). The student's knowledge uncertainty represents the inherent uncertainty in the student model's knowledge, derived using Equation 5 in the paper. The key insight of our work is that both deterministic knowledge and uncertain knowledge from the teacher should be simultaneously considered during the knowledge distillation process. Naturally, the student model also can capture and leverage both types of knowledge. Therefore, we conducted specific experiments to explore the student's uncertain knowledge, as presented in Table 6.

---

> > ### Comment · Reviewer_pHpX · 2024-11-25
> >
> > Thanks for the response, which partially answer my questions. I greatly appreciate the authors’ effort in providing additional KD results on other detectors and KD approaches. However,
> >
> > 1. Regarding uncertain knowledge, UET is relatively similar to AKD. The authors do not provide relevant discussions to show the difference between UET and AKD. Empirically, I would be looking for significant mAP gains of UET without deterministic knowledge (i.e., without the original teacher), which may better reflect the advantages of UET over AKD.
> >
> > 2. Given the point raised in 1, compared with AKD, the superiority of UET is actually attributed to deterministic knowledge, i.e., the original teacher. This causes UET to be more like an incremental work. The simple combination (key insight of UET, as the authors stated) of uncertain and deterministic knowledge for distillation may not sufficiently underpin the novelty of this work. To be clear, as recognized by Reviewer kB7D and extensively investigated in the past, multi-teacher KD is bound to deliver better KD results.
> >
> > I have also gone through the other reviewers’ feedback and authors’ responses to them. Based on all these, I will maintain my original recommendation of 5.

---

> ### Author Response · Authors · 2024-11-25
>
> Thank you for the new feedback.
>
> Indeed, uncertainty is not a new idea given [1]  has 5,743 citations.  To our knowledge, PAD [2] is the first approach to applying the concept of uncertainty to knowledge distillation, and AKD [3] builds upon PAD as an enhanced version. For clarity, we list their objectives below:
>
> $
> \mathcal{L}\_{PAD} = \sum\_{i=1}^{N} \left( \frac{\left(f\_s(x\_i) - y\_i\right)^2}{\sigma\_i^2} + \ln\sigma\_i^2 \right).
> $  (PAD Eq.7, 2020)
>
> $
> \mathcal{L}\_{AKD}(S, A) = \frac{1}{k} \sum\_{i=1}^{k} \frac{1}{HWC}
> \left\|
> \frac{F^{(a\_i)}\_{c,h,w}}{\sigma\_{c,h,w}} - \frac{\phi(F^{(s)})\_{c,h,w}}{\sigma\_{c,h,w}}
> \right\|\_2^2
> = \frac{1}{k \cdot HWC} \sum\_{i=1}^{k} \sum\_{c,h,w}
> \frac{\left(F^{(a\_i)}\_{c,h,w} - \phi(F^{(s)})\_{c,h,w}\right)^2}{\sigma\_{c,h,w}^2}.
> $ (AKD Eq.7, 2023)
>
> For your clarity, we will include the discussion of PAD in the manuscript.
>
> Generally, they model the uncertainty as the variance of the feature space. In contrast, our approach introduces MC Dropout to measure the uncertainty of knowledge. We know this scheme is embarrassingly simple yet we think it is a strength with soundness, as also denoted by Reviewer kB7D.
>
> We sincerely hope this explanation addresses your confusion between the concepts of uncertainty and KD. Thank you again for your valuable feedback.
>
> [1] What uncertainties do we need in Bayesian deep learning for computer vision? NeurIPS 2017
>
> [2] Prime-aware adaptive distillation. ECCV 2020
>
> [3] Avatar knowledge distillation: self-ensemble teacher paradigm with uncertainty. ACM MM 2023

---

> > ### Comment · Reviewer_pHpX · 2024-12-03
> >
> > Sorry for the late update. I think there are some misunderstandings.
> >
> > 1.	I totally agree that dropout is a simple yet effective scheme for measuring knowledge uncertainty. This is a strength indeed. My concern is that UET and AKD share the idea of generating uncertain knowledge with the dropout for distilling detectors. In other words, you only need to clarify the essential differences between UET and AKD regarding how to estimate and leverage knowledge uncertainty in KD. It would be better if UET (w/o deterministic knowledge) also has significant performance advantages over AKD.
> >
> > 2.	If the concern in 1 is valid, combining the uncertain knowledge (already explored in AKD) and deterministic knowledge (the common practice in KD) for distillation is an incremental work, from where I stand.
> >
> > A kind reminder: using NeurIPS instead of NIPS, to avoid unintentional issues.

---

> > > ### Author Response · Authors · 2024-12-03
> > > **Clarification on Dropout Usage and Uncertain Knowledge**
> > >
> > > We sincerely appreciate your valuable comments. To clarify, the dropout mechanisms in AKD and UET serve fundamentally different purposes:
> > >
> > > 1). Dropout Mechanism
> > >
> > > AKD uses dropout as a replaceable perturbation to introduce diversity, while UET employs MC Dropout [1], a well-established uncertainty estimation method that is irreplaceable in its theoretical grounding.
> > >
> > > 2). Uncertain Knowledge
> > >
> > > AKD generates noise knowledge (not uncertain knowledge) through dropout and relies on variance modeling to approximate uncertainty and mitigate noise. In contrast, UET directly uses MC Dropout to produce true uncertain knowledge, which is inherently transferable and integral to the distillation process.
> > >
> > > Thank you again for your constructive feedback.
> > >
> > > [1] What uncertainties do we need in Bayesian deep learning for computer vision? NeurIPS 2017.

---

### Author Response · Authors · 2024-11-21

We thank the AC and all reviewers for their valuable time and constructive comments. We are encouraged by the reviewers' acknowledgments regarding the novelty of our UET paradigm and its effectiveness in enhancing knowledge distillation (Reviewer kB7D and mLFV). We appreciate the reviewers' praise for the paper being well-organized, well-written, and easy to follow (Reviewer kB7D). We also thank the reviewers for recognizing the proposed method's effectiveness and improvements (Reviewer pHpX, kB7D, and mLFV).
Below, we provide detailed responses to the main points raised:

1. **Comparisons with Transformer-Based Models**

We have extended our experiments to include Transformer-to-Transformer configurations. Specifically, we used Swin Transformer-tiny as the teacher backbone and Swin Transformer-nano as the student backbone. The results are shown in follows table. When employing PKD for distillation, the student network achieves an mAP of 32.2. By integrating the proposed UET module, the mAP of the student model is further improved, reaching 32.5. These findings further confirm the generalization capability of UET in Transformer-based settings.

| Methods                 | Schedule | mAP          | $AP_{50}$ | $AP_{75}$ | $AP_S$ | $AP_M$ | $AP_L$ |
|-------------------------|----------|--------------|-----------|-----------|--------|--------|--------|
| SwinT-tiny (T)          | 1$\times$ | 37.3         | 57.5      | 39.9      | 22.7   | 41.0   | 49.6   |
| SwinT-nano (S)          | 1$\times$ | 31.4         | 50.5      | 32.9      | 18.0   | 34.0   | 41.5   |
| PKD                     | 1$\times$ | 32.2 (+0.8)  | 50.5      | 34.2      | 17.2   | 34.9   | 45.3   |
| **PKD+Ours**            | 1$\times$ | **32.5 (+1.1)** | 50.8      | 34.4      | 18.6   | 35.2   | 45.5   |

2. **Generalization to different KDs on RetinaNet detector**

We added experiments embedding UET into multiple KD methods (MGD and PKD) on the RetinaNet detector to evaluate the generalization ability further. The results are presented in the following Table. When using MGD for deterministic knowledge distillation, the student model achieves a performance of 39.3 mAP, representing a 1.9 improvement over the original student model. By adapting MGD to follow the UET paradigm, the student model's performance further increases to 39.6 mAP, achieving a total improvement of 2.2 points over the original student.
Similarly, with PKD, the student model achieves 39.6 mAP, which is a 2.2 improvement over the original student. When PKD is adapted to the UET paradigm, the student model's performance further improves to 39.8 mAP, resulting in a total improvement of 2.4 points over the original student.

These extensive experiments demonstrate that UET is a versatile and robust framework, capable of improving various KD methods across multiple detectors and configurations.

| Method                 | Schedule      | mAP           | $AP_{50}$ | $AP_{75}$ | $AP_S$ | $AP_M$ | $AP_L$ |
|------------------------|---------------|---------------|-----------|-----------|--------|--------|--------|
| RetinaNet-Res101 (T)   | 2$\times$, ms | 38.9          | 58.0      | 41.5      | 21.0   | 42.8   | 52.4   |
| RetinaNet-Res50 (S)    | 1$\times$     | 37.4          | 56.7      | 39.6      | 20.6   | 40.7   | 49.7   |
| MGD*                   | 1$\times$     | 39.3 (+1.9)   | 58.6      | 41.9      | 22.3   | 43.2   | 52.3   |
| **MGD+Ours**           | 1$\times$     | **39.6 (+2.2)** | 58.6      | 42.5      | 22.3   | 43.8   | 52.6   |
| PKD*                   | 1$\times$     | 39.6 (+2.2)   | 58.8      | 42.7      | 22.3   | 43.8   | 54.1   |
| **PKD+Ours**           | 1$\times$     | **39.8 (+2.4)** | 58.8      | 42.6      | 22.2   | 43.9   | 53.9   |

---

> ### Author Response · Authors · 2024-11-21
>
> 3. **More Comparison with AKD**
>
> We perform additional comparison experiments with AKD. The results, summarized in the following Table, demonstrate the consistent superiority of our method across the Faster R-CNN and FCOS detectors. Specifically, our approach achieves an mAP of 40.8 on Faster R-CNN, surpassing AKD's 40.6, and 42.9 on FCOS, outperforming AKD's 42.7. These results highlight the effectiveness of UET in enhancing the KD process.
>
> | Method                          | Schedule      | mAP   | $AP_{50}$ | $AP_{75}$ | $AP_S$ | $AP_M$ | $AP_L$ |
> |---------------------------------|---------------|-------|-----------|-----------|--------|--------|--------|
> | **Two-stage detectors**         |               |       |           |           |        |        |        |
> | Faster R-CNN-Res101 (T)         | 2$\times$     | 39.8  | 60.1      | 43.3      | 22.5   | 43.6   | 52.8   |
> | Faster R-CNN-Res50 (S)          | 2$\times$     | 38.4  | 59.0      | 42.0      | 21.5   | 42.1   | 50.3   |
> | FGD                             | 2$\times$     | 40.5  | -         | -         | 22.6   | 44.7   | 53.2   |
> | +AKD                            | 2$\times$     | 40.6  | 60.2      | 43.9      | 22.8   | 44.3   | 53.6   |
> | **+Ours**                       | 2$\times$     | **40.8** | **61.0** | **44.5** | **23.5** | **44.9** | **53.7** |
> | **One-stage detectors**         |               |       |           |           |        |        |        |
> | FCOS-Res101 (T)                 | 2$\times$, ms | 40.8  | 60.0      | 44.0      | 24.2   | 44.3   | 52.4   |
> | FCOS-Res50 (S)                  | 2$\times$, ms | 38.5  | 57.7      | 41.0      | 21.9   | 42.8   | 48.6   |
> | FGD                             | 2$\times$     | 42.7  | -         | -         | **27.2** | 46.5   | **55.5** |
> | +AKD                            | 2$\times$     | 42.7  | 61.4      | 46.1      | 26.9   | 46.5   | 54.6   |
> | **+Ours**                       | 2$\times$     | **42.9** | **61.6** | **46.3** | 27.1   | **46.8** | 54.7   |

---

### Meta-Review · Area_Chair_5mVk · 2024-12-19

**Metareview:**

(a) This paper proposes UET, a method for improving knowledge distillation in object detection by leveraging MC dropout to estimate teacher uncertainty, enabling diverse knowledge transfer, and demonstrates its effectiveness through comprehensive experiments.

(b) Strengths: The paper presents a convincing motivation by highlighting the benefits of multiple teachers providing diverse and informative supervision to students. UET is seamlessly integrated with existing knowledge distillation (KD) methods and achieves state-of-the-art performance. It extends successfully to classification and semantic segmentation tasks, demonstrating versatility across domains. The method shows strong empirical results, outperforming other methods across various detectors, including one-stage and two-stage models, with results on smaller backbones and additional tasks provided for completeness. The writing is clear and allows easy reproducibility.

(c) The weaknesses of the initial submission highlight concerns about novelty, as UET's idea is similar to prior work (AKD), though an ablation study on this aspect is missing. Generalization remains underexplored, as UET is not extensively evaluated across different detectors and KD methods, and comparisons with Transformer-based models are absent. Some results show inconsistencies, and the paper lacks clarity on specific details. Additionally, the use of Monte Carlo dropout for sampling may slow down the training process.

(d) The most important reasons for the AC's decision to reject are: the proposed UET indeed has not significant differences over AKD. There are two reviewers (pHpX, mLFV) engaged in the discussions about the issue. First of all, the AC appreciates that the authors clarify clearly the key differences between UET and AKD: 1) dropout formulation; 2) complementary knowledge integration vs. uncertainty weighting. The AC carefully check the discussions and related papers: From the perspective of the algorithm pipeline, there is indeed not much difference between the two works. The process consists of two steps: the first is feature sampling, and the second is uncertainty-driven distillation. The proposed UET and the related AKD share the same pipeline but just differ in the design details in each step, e.g., MC Dropout vs. perturbation Dropout. That's also the reason why Reviewer pHpX believes that UET has no more significant differences over AKD. Further, from Tabel 3 and 5, the AC notices that the performance of AKD is close to that of UET (43.9 vs. 44.1), and the hyper-parameter could also lead to the performance gap > 0.2 (e.g., M = 15, 43.8; M = 10, 44.1). The optimal performance of UET (44.1) appears to result from careful tuning, particularly with respect to M. However, it is unclear whether the performance of AKD has been similarly optimized through its hyperparameters (the AC fails to find corresponding descriptions about this). It could be unfair as the authors highlight the performance gain of 0.2 over AKD in the revised manuscript.

**Additional Comments On Reviewer Discussion:**

(a) Reviewer pHpX points out that the paper's novelty is weakened by its similarity to AKD, with the primary distinction being the inclusion of the original teacher in UET, but no ablation study evaluates its impact. They express concerns about fairness, as the superior results rely heavily on FGD, making comparisons with SOTA methods less convincing. The reviewer also highlights insufficient generalization tests across detectors and KD methods and seeks clarification on the student's knowledge uncertainty (denoted as S) in Table 6. The authors have successfully most of the concerns, but the reviewer still feels that UET has no more significant differences over AKD. The AC agrees with this point from the perspective of algorithm pipeline.

(b) Reviewer kB7D identifies some writing issues and notes inconsistencies in the reported results despite their initial strength. They highlight the lack of comparisons with Transformer models and insufficient reporting on calibration metrics. Additionally, the reviewer points out the omission of timing performance evaluation. The authors address these issues, especially correct some reported performance regarding FGD results.

(c) Reviewer mLFV notes that UET shows only a modest improvement of 0.2 mAP over the LD-only model in Table 7 and questions why logit-based KD methods benefit less from UET. They suggest that having the student generate and imitate uncertainty features directly might be a more suitable approach. Additionally, they point out that the Monte Carlo dropout, requiring multiple sampling iterations, may slow down the training process. The authors carefully address all the concerns. Further, the reviewer engages in the discussion regarding the differences with AKD, he concludes that "concern about the difference between UET and AKD has been partly solved" given the feedbacks of the authors.

---

### Decision · Program_Chairs · 2025-01-22

Reject